# Effects of Reanalysis Forcing Fields on Ozone Trends and Age-of-Air from a Chemical Transport Model

Yajuan Li[1], Sandip S. Dhomse[2,3], Martyn P. Chipperfield[2,3], Wuhu Feng[2,4], Andreas Chrysanthou[2], Yuan Xia[1] and Dong Guo[5]

[1]School of Electronic Engineering, Nanjing Xiaozhuang University, Nanjing, China
[2]School of Earth and Environment, University of Leeds, Leeds, UK
[3]National Centre for Earth Observation, University of Leeds, Leeds, UK
[4]National Centre for Atmospheric Science, University of Leeds, Leeds, UK
[5]Key Laboratory of Meteorological Disaster, Ministry of Education/Joint International Research Laboratory of Climate and Environment Change/Collaborative Innovation Center on Forecast and Evaluation of Meteorological Disasters, Nanjing University of Information Science & Technology, Nanjing, China

*Correspondence to*: Yajuan Li (yajuanli@njxzc.edu.cn), Sandip Dhomse (S.S.Dhomse@leeds.ac.uk)

**Abstract.** We use the TOMCAT, a 3-dimensional (3D) off-line chemical transport model (CTM) forced by two different meteorological reanalysis datasets (ERA-Interim and ERA5) from the European Centre for Medium-Range weather Forecasts (ECMWF) to analyse seasonal behaviour and long-term trends in stratospheric ozone and mean age-of-air. The model-simulated ozone variations are evaluated against two observation-based data sets. For total column ozone (TCO) comparisons we use the Copernicus Climate Change Service (C3S) data (1979-2019), while for ozone profiles we use the Stratospheric Water and OzOne Satellite Homogenized (SWOOSH) dataset (1984-2019). We find that the CTM simulations forced by ERA-Interim (A_ERAI) and ERA5 (B_ERA5) can both successfully reproduce spatial and temporal variations in stratospheric ozone. Also, modelled TCO anomalies from B_ERA5 show better agreement with C3S than A_ERAI, especially in northern hemisphere (NH) mid-latitudes, except that it gives somewhat larger positive biases (> 15 DU) during winter-spring seasons. Ozone profile comparisons against SWOOSH data show larger differences between the two simulations. In the lower stratosphere, ozone differences can be directly attributed to the representation of dynamical processes, whereas in the upper stratosphere they can be directly linked to the differences in temperatures between ERAI and ERA5 data sets. Although TCO anomalies from B_ERA5 show relatively better agreement with C3S compared to A_ERAI, comparison with SWOOSH data does not confirm that B_ERA5 performs better in simulating the variations in the stratospheric ozone profiles. We employ a multi-variate regression model to quantify the TCO and ozone profile trends before and after peak stratospheric halogen loading in 1997. Our results show that compared to C3S, TCO recovery trends (since 1998) in simulation B_ERA5 are significantly overestimated in the southern hemisphere (SH) mid-latitudes, while for A_ERAI in the NH mid-latitudes simulated ozone trends remain negative. Similarly, in the lower stratosphere B_ERA5 shows positive ozone recovery trends for both NH and SH mid-latitudes. In contrast, both SWOOSH and A_ERAI show opposite (negative) trends in the NH mid-latitudes.

Furthermore, we analyse age-of-air (AoA) trends to diagnose transport differences between the two reanalysis data sets. Simulation B_ERA5 shows a positive AoA trend after 1998 and somewhat older age in the NH lower stratosphere compared to A_ERAI, indicating a slower Brewer-Dobson circulation does not translate into reduced wintertime ozone build-up in the NH extratropical lower stratosphere. Overall, our results show that models forced by the most recent ERA5 reanalyses may not yet be capable of reproducing observed changes in stratospheric ozone, particularly in the lower stratosphere.

## 1 Introduction

The stratospheric ozone layer protects life on earth from the damaging effects of ultraviolet radiation. The 1987 Montreal Protocol and its subsequent amendments and adjustments have successfully controlled the major anthropogenic ozone-depleting substances (ODSs) leading to a decrease in stratospheric chlorine and bromine and the onset of recovery of the ozone layer (e.g. WMO, 2018). The characteristic details of ozone depletion and the ongoing recovery in recent decades has been investigated using both observations and models (e.g. Solomon et al., 2016; Chipperfield et al., 2017; Dhomse et al., 2018; WMO, 2018 and references therein).

Previous studies consistently report a robust sign of recovery in upper stratospheric ozone after the peak halogen (chlorine and bromine) loading around the year 1997 (e.g. Chipperfield et al., 2017; Weber et al., 2018). Besides the decrease in ODSs, the increased greenhouse gas (GHG) abundances warm the troposphere, causing strengthening of the stratospheric circulation and increases in tropical upwelling, which reduces ozone in the tropical lower stratosphere (e.g. Bekki et al., 2013; Marsh et al., 2016). Also, increasing GHGs cause stratospheric cooling that slows gas-phase ozone loss cycles and is expected to speed-up recovery in upper stratospheric ozone globally (e.g. Haigh and Pyle, 1982; Eyring et al., 2010; Douglass et al., 2012). However, the recovery of ozone in the upper stratosphere does not imply the recovery of the stratospheric or whole atmosphere column ozone. In the lower stratosphere, a region characterised by large interannual variability, the evolution of ozone is much more complicated as its abundance is largely controlled by complex interactions between various chemical and dynamical processes (e.g. WMO, 2014). Even with those complications, it was expected that first signs of ozone recovery (i.e. almost negligible negative ozone trends) would be detectable within a couple of decades after the peak in stratospheric chlorine loading. However, recent observation-based studies show evidence of a continued decline in lower stratospheric ozone since 1998 (e.g. Steinbrecht et al., 2017; Ball et al., 2018, 2019).

Using model simulations, dynamical variability has been proposed as the possible driver that dominates the recent ozone changes in the mid-latitude lower stratosphere (e.g. Chipperfield et al., 2018; Stone et al., 2018). However, inconsistencies have been noted between the observed and model-simulated ozone variations. Ball et al. (2018) reported a significant decrease in lower stratospheric ozone between 60°S and 60°N over the period 1998-2016 using multiple satellite datasets. Furthermore, there was no significant change in total column ozone due to cancellation of opposing trends from increasing tropospheric ozone. They also compared stratospheric partial column ozone trends with two chemistry–climate models (CCMs) run in a "specified-dynamics" configuration constrained with reanalyses, neither of which reproduced the observed

lower stratospheric decline, possibly related to limitations in capturing the residual circulation adequately (e.g. Chrysanthou et al., 2019; Orbe et al., 2020a). Subsequently, the negative trends in the mid-latitude lower stratospheric ozone have been identified from reanalysis results and updated satellite datasets (e.g. Wargan et al., 2018; Ball et al., 2019). Chipperfield et al. (2018) demonstrated the ability of TOMCAT/SLIMCAT chemical transport model (CTM) simulations to largely reproduce the observed ozone changes and suggested that atmospheric dynamics plays an important role in controlling ozone in the

extra-polar lower stratosphere. They also showed that the effects of trends in short-lived chlorine and bromine compounds on the recent ozone changes are relatively small. Ball et al. (2019) extended their analysis through 2018 and proposed that the global lower stratospheric ozone decrease is continuing despite the large, short-term ozone increase in 2017, which might have been overestimated in CTM simulations by Chipperfield et al. (2018).

    Orbe et al. (2020b) showed that a free-running CCM can simulate the ozone decrease in the northern hemisphere lower

stratosphere, but the magnitude of ozone changes is significantly weaker than observed, and consistent with weaker residual circulation changes. Ball et al. (2020) also showed that CCMVal models run with a future ODS and GHG scenario (REF-C2) exhibit a decline in tropical lower stratospheric ozone similar to that observed, but most CCMs do not reproduce the observed decrease in the mid-latitude lower stratosphere. Dietmüller et al. (2021) recently investigated a set of 31 CCM simulations and found that none of the model simulations reproduces the coherent negative ozone trends in the tropical and

extra-tropical lower stratosphere as suggested by recent observational data sets. Instead, most simulations show a dipole pattern with the tropical ozone trend opposite to that in mid-latitudes. These inconsistencies between model simulations and observations imply that dynamical effects on the lower stratospheric ozone changes are still not well understood.

    Chemical transport model simulations are ideally suited for interpreting the past ozone changes as well as for quantifying the influence of important physical processes on the ozone variability. However, model-simulated ozone distributions

generally show some biases with respect to observation-based datasets due to uncertain photochemical parameters, transport errors and other simplifications of computationally expensive processes (e.g. WMO, 2014, 2018; Dhomse et al., 2018, 2021). The inability of chemical models to simulate the observed lower stratospheric ozone decrease can be largely attributed to the model deficiencies in, for example, transport (Chipperfield et al., 2018; Ball et al., 2018, 2020). Additionally, most observational data records also show large errors due to the measurement technique, instrument limitations or degradation

(e.g. Hubert et al., 2016; SPARC, 2019). Hence, comparison between observations and model simulations generally shows time-varying differences. An increase in vertical resolution as well as inclusion of complex chemical and dynamical processes is generally recommended to reduce biases in model-simulated ozone (e.g. Feng et al., 2011; Dhomse et al., 2011).

    As CTMs are forced with (re)analysis meteorological fields they are better suited to understand past ozone changes compared to free-running CCMs. Over the time, improvements achieved in meteorological reanalyses such as those from the

European Centre for Medium-Range Weather Forecasts (ECMWF) have led to the better representation of stratospheric transport (e.g. Monge-Sanz et al., 2013; Diallo et al., 2021). With the ECMWF fifth generation reanalysis ERA5 (Hersbach et al., 2020) superseding ERA-Interim (Dee et al., 2011), a key question is whether the new reanalysis can improve the simulation performance with respect to the older one when it is used to force CTM simulations (Albergel et al., 2018). It

should be noted that there could be inhomogeneities in reanalysis datasets due to changes in available observations assimilated as well as instrument degradation that could introduce spurious transport features (e.g. Schoeberl et al., 2003; Ploeger et al., 2015). Here, we focus on the model performance in interpreting key characteristics of stratospheric ozone using CTM simulations forced by ECMWF ERA-Interim and ERA5 reanalysis datasets. By comparing with observation-based data sets, we evaluate the quality of model simulations and investigate possible reasons for their differences.

The paper is organized as follows. Section 2 describes the CTM simulations forced by ERA-Interim and ERA5 reanalyses, followed by the satellite datasets and regression methods. Section 3 compares the variability and trends in ozone total column and vertical profiles between simulations and observations. The mean age-of-air distributions and their trends are compared and discussed in Section 4, followed by our summary and conclusions in Section 5.

## 2 Data and methods

### 2.1 Model and simulations

Here we use the global off-line 3-D CTM (TOMCAT/SLIMCAT, hereafter TOMCAT) which has been described in detail by Chipperfield (2006). The model contains a detailed description of stratospheric chemistry (e.g. Feng et al., 2011, 2021; Chipperfield et al., 2018), including the concentrations of major ODSs and GHGs, aerosol effects from volcanic eruptions, and variations in solar forcing. For major ODSs and GHGs, the model uses updated global mean surface mixing ratio scenarios (Carpenter et al., 2018) which are treated as well mixed throughout the troposphere. The implementation of sulphate aerosol surface area density (SAD) and solar flux variations are described in Dhomse et al. (2015; 2016). Aerosol (or SAD) variations from volcanic eruptions are from Luo (2016), whereas solar flux variations are taken from the NRL2 empirical model (Coddington et al., 2016).

ECMWF ERA-Interim reanalyses have been extensively used to drive CTM simulations for multi-annual trend investigations (e.g. Chipperfield et al., 2017; Feng et al., 2021). These reanalyses are based on a coherent assimilation of observations using an atmospheric general circulation model (Dee et al., 2011), covering the period from January 1979 to August 2019. ERA5 is the latest reanalysis product released by ECMWF, to supersede ERA-Interim, and comprehensive account is provided by Hersbach et al. (2020). Both ERA5 and ERA-Interim apply 4-dimensional variational analysis (4D-Var). ERA5 resolves the atmosphere using 137 levels from the surface up to 0.01 hPa (~80 km) with a horizontal spatial resolution of 31 km, while ERA-Interim uses 60 levels from the surface to 0.1 hPa (~65 km) and 80 km for horizontal resolution. Although almost all of the radiance datasets assimilated in ERA-Interim are included in ERA5, an updated radiative transfer model is used in ERA5 (RTTOVS-v11 against RTOVS v7) and it includes several developments and various reprocessed radiance datasets (see Figure 5 in Hersbach et al., 2020). Major differences between two reanalysis datasets also include significant divergence in terms of volume of radiance measurements assimilated post-2007 (more and newer observations in ERA5), which are not assimilated in ERA-Interim, together with the gradual decline in the numbers assimilated in ERA-Interim, as instruments and channels gradually failed. In both ERA-Interim and ERA5 reanalysis

systems, the prognostic ozone model is based on the parameterization scheme of Cariolle and Teyssèdre (2007). Simply, both systems have the ozone evolution that is expressed as a linear expansion with respect to the photochemical equilibrium for the local values of the ozone mass mixing ratio, the overhead ozone column, the temperature and some ozone depletion during polar winter. The ozone tracer is advected with the model flow. It is important to note that the ERA5 prognostic ozone is not coupled with the radiation scheme which uses diagnostic time varying ozone fields recommended for CMIP5 simulations.

Here we perform two TOMCAT simulations, A_ERAI and B_ERA5, which are forced with ERA-Interim and ERA5 reanalysis datasets (Dhomse et al., 2019; Feng et al., 2021), respectively. The simulations use identical chemical and dynamical parameters for the whole time period available in ERA-Interim from January 1979 to August 2019. Simulation B_ERA5 uses the corrected ERA5.1 analyses for the period from 2000 to 2006; these have better global-mean temperatures in the lower stratosphere than provided by the original ERA5 product (Simmons et al., 2020). The model grid is variable and determined by converting the forcing ECMWF meteorological analyses to grid-point fields using a spectral transformation (Chipperfield, 2006). Both TOMCAT simulations are performed at 2.8° × 2.8° (T42 Gaussian grid) horizontal resolution and have 32 hybrid sigma-pressure levels ranging from the surface to about 60 km. The 6-hourly grid point meteorological fields are interpolated linearly in time for both runs. Although ERA5 provides hourly output including information about uncertainties, here we used 6-hourly fields as ERA-Interim provides output at the same frequency and to reduce storage requirements.

## 2.2 Observation-based datasets

We use the total column ozone (TCO) data from the Copernicus Climate Change Service (hereafter C3S, obtained from https://cds.climate.copernicus.eu/cdsapp#!/dataset/satellite-ozone-v1?tab=overview, last access: May 2022) for quantification of long-term variability and trends. This monthly mean gridded dataset is created by combining total ozone data from 15 satellite sensors, including the Global Ozone Monitoring Experiment (GOME, 1995-2011), Scanning Imaging Absorption Spectrometer for Atmospheric CHartographY (SCIAMACHY, 2002-2012), Ozone Monitoring Instrument (OMI, 2004-present), GOME-2A/B (2007-present), Backscatter Ultraviolet Radiometer (BUV-Nimbus4, 1970-1980), Total Ozone Mapping Spectrometer (TOMS-EP, 1996-2006), Solar Backscatter Ultraviolet Radiometer (SBUV-9, -11, -14, -16, -17, -18, -19, 1985-present) and Ozone Mapping and Profiler Suite (OMPS, 2012-present). This merged product spans from 1970 to present and the horizontal resolution after January 1979 is 0.5° × 0.5°. The long-term stability of the total column product is within the 1%/decade level. Systematic and random errors in this data are below 2% and 3-4%, respectively, which makes it suited for long-term trend analysis. Detailed validation of C3S total column ozone data is available at https://datastore.copernicus-climate.eu/documents/satellite-ozone/C3S2_312a_Lot2_PQAR_O3_latest.pdf (last access: June 2022). Li et al. (2020b) also showed that there is no long-term drift in the C3S data and found that the differences between C3S and the SBUV satellite data are less than 2-3% throughout the record 1979-2017.

The Stratospheric Water and OzOne Satellite Homogenized (SWOOSH, obtained from https://csl.noaa.gov/ groups/csl8/swoosh/, last access: May 2022) dataset is used to evaluate our simulated ozone profiles. SWOOSH (v2.7) includes a merged record of stratospheric ozone and water vapour measurements, comprised of data from the Stratospheric Aerosol and Gas Experiment (SAGE-II/III), Upper Atmospheric Research Satellite Halogen Occultation Experiment (UARS HALOE), UARS Microwave Limb Sounder (MLS), and Aura MLS instruments (Davis et al., 2016 and references therein). The measured values are homogenized by applying the corrections calculated from data collected during the overlapping time periods of the instrument. The merged SWOOSH record spans from 1984 to present, and consists of monthly mean zonal-mean ozone values at grids of 2.5° and 12 levels per decade in pressure from 316 to 1 hPa (31 pressure levels). Comparisons between the SWOOSH merged product and independent ground-based measurements (e.g. Hubert et al., 2016) and satellite data sets (e.g. Harris et al., 2015) confirm the long-term stability of the SWOOSH ozone product.

The stratospheric mean age, as a pure transport diagnostic, is calculated from measurements of a chemically inert tracer gas whose concentration growth varies linearly with time (e.g. Hall and Plumb, 1994; Waugh and Hall, 2002; Chipperfield, 2006). The annually averaged $CO_2$ and $SF_6$ are two tracers widely used for mean age estimates. As shown in Figure 5 of Hall et al. (1999), there is good agreement in mean age as inferred from different in situ $CO_2$ and $SF_6$ measurements (e.g. Boering et al., 1996; Elkins et al., 1996). These in situ measurements were carried out during the period from 1992 to 1997. Here, we use them as a benchmark to evaluate the impact of meteorological variability on the modelled mean age-of-air (AoA).

## 2.3 Regression methods

Multi-variate linear regression models (MLR) are widely used to assess long-term ozone trends (e.g. Solomon et al., 1996; Reinsel et al., 2002; Dhomse et al., 2006; Randel and Wu, 2007; Fioletov, 2009; Steinbrecht et al., 2017; Li et al., 2020b). Generally, the piecewise linear trends (PLWT), also called hockey stick (Harris et al., 2008), and the equivalent effective stratospheric chlorine (EESC) term used in MLR are applied to determine the ozone trends (e.g. Chehade et al., 2014). However, in order to avoid complications arising from fitting of the second trend term, we use the independent linear trends (ILT, Weber et al., 2018) to consider the trends before and after the turnaround of the peak stratospheric halogen loading. The MLR equation with ILT terms is given by:

$$Y(t) = a_1 \cdot A_1(t) + b_1 \cdot A_1(t) \cdot (t_0 - t) + a_2 \cdot A_2(t) + b_2 \cdot A_2(t) \cdot (t - t_0) + \sum c_i \cdot X_i(t) + \varepsilon(t) \qquad (1)$$

where Y(t) is the ozone time series for the total column ozone or the vertically resolved ozone. The coefficients $b_1$ and $b_2$ are the linear trends before and after the turnaround year ($t_0$=1997). $a_1$ and $a_2$ are intercepts which make the two linear trends discontinuous and independent of each other. The multiplication with $A_1(t)$ and $A_2(t)$ describes mathematically the periods before and after the turnaround year. $A_1(t)$ and $A_2(t)$ are given by

$$A_1(t) = \begin{cases} 1, & t \le t_0 \\ 0, & t > t_0 \end{cases}$$

and

$$A_2(t) = \begin{cases} 0, & t \le t_0 \\ 1, & t > t_0 \end{cases}$$

respectively. $X_i(t)$ stands for explanatory proxies, including the solar flux for the 11-year solar cycle (obtained from http://www.iup.uni-bremen.de/UVSAT/Datasets/mgii, last access: May 2022), the quasi-biennial oscillation (QBO) at 30 hPa and 10 hPa (QBO30 and QBO10, http://www.geo.fuberlin.de/met/ag/strat/produkte/qbo/qbo.dat, last access: May 2022), stratospheric aerosol loading from volcanic eruptions (https://data.giss.nasa.gov/modelforce/strataer/, last access: May 2022), El-Nino Southern Oscillation (ENSO), Arctic Oscillation (AO) or Antarctic Oscillation (AAO) index. ENSO, AO and AAO indices are obtained from Climate Prediction Centre via https://www.cpc.ncep.noaa.gov/ (last access: May 2022). $c_i$ represents the time-dependent regression coefficient of each proxy $X_i$ and $\varepsilon$ is the residual term. All these explanatory proxies are de-trended (except for the aerosol term) and normalised between 0 and 1. Correlation analysis is applied to ensure that explanatory proxies used in MLR should not be highly correlated with each other.

The regression model used for the total column ozone in December-January-February (DJF) and June-July-August (JJA) seasons are identical to that used in Li et al. (2020b). The proxies are also averaged for DJF and JJA seasons. For the vertically resolved monthly mean ozone profiles, seasonal components consisting of sinusoidal terms of periods 12, 6, 4, and 3 months are also considered in the MLR. Meanwhile the Cochrane-Orcutt transformation with a time lag of one month is applied to the regression equation to avoid non-negligible auto-correlation in the residuals (e.g. Cochrane and Orcutt, 1949; Reinsel et al., 2002; Dhomse et al., 2006). The correlation analysis for the DJF mean and monthly mean explanatory variables are shown in the supplementary **Table S1** and **Table S2**. As the correlation coefficients between some proxies (e.g. solar and AO, ENSO and aerosol, AAO and aerosol) are significant, we simply consider ILT terms and traditional proxies (solar, QBO, ENSO) in the MLR. The ozone values for the four years of 1982, 1983, 1991 and 1992 are removed as the aerosol term is excluded from the MLR. The ozone trend profiles from 147 hPa to 1 hPa (100 hPa to 1 hPa for the tropical region) are calculated with the coefficients referenced to the ozone values at different pressure levels.

# 3 Results

## 3.1 Variability and trends in total column ozone

### 3.1.1 Total column ozone anomalies

To evaluate the performance of model simulations compared to the observations, we first look at the characteristics of total column ozone (TCO) anomalies in different latitude regions over the extended time period 1979-2019 (August). Anomalies are calculated by subtracting the long-term monthly average from each monthly mean value. **Figures 1a-e** (left column) show the monthly mean TCO anomalies obtained from C3S and TOMCAT simulations, A_ERAI and B_ERA5, over 1979-2019 (August) for the NH high-latitudes (90°N-60°N), mid-latitudes (60°N-35°N), tropics (20°N-20°S), SH mid-latitudes (35°S-60°S) and SH high-latitudes (60°S-90°S). The absolute differences of the climatological anomalies between each simulation and C3S, as well as between the two model simulations (B_ERA5 - A_ERAI), are also shown in **Figures 1f-j** (right column). Overall, both model simulations are able to capture the temporal characteristics in ozone variations relative to C3S very well, confirming the realistic representation of important chemical and dynamical processes in TOMCAT. However, the magnitude and structure of the inter-annual total ozone anomalies show different aspects of differences between two reanalysis data sets in different latitude regions. For example, correlation analysis between simulated and C3S TCO anomalies shows that B_ERA5 is better correlated to C3S than A_ERAI for most latitude regions. In particular, in the NH mid-latitude region B_ERA5 shows much better correlation (0.93) with C3S than A_ERAI (0.79), meaning that B_ERA5 anomalies track observed anomalies better than A_ERAI, especially during 1980s. An interesting feature in **Figures 1f-g** is that simulations A_ERAI and B_ERA5 show significant differences at NH mid and high latitudes. The comparison also shows that before 1998 anomalies from B_ERA5 are relatively smaller than from A_ERAI (up to ~ -20 DU biases – shaded green regions) but are larger during later years (up to ~ +20 DU biases – shaded yellow regions). The relatively better agreement between B_ERA5 and C3S in the NH mid-high-latitude regions could be due to improvements in the representation of dynamical processes in ERA5 reanalysis data, such as convection in the upper troposphere and lower stratosphere (Li et al., 2020a) and residual mean mass circulation of the BDC in the stratosphere (Diallo et al., 2021; Ploeger et al., 2021).

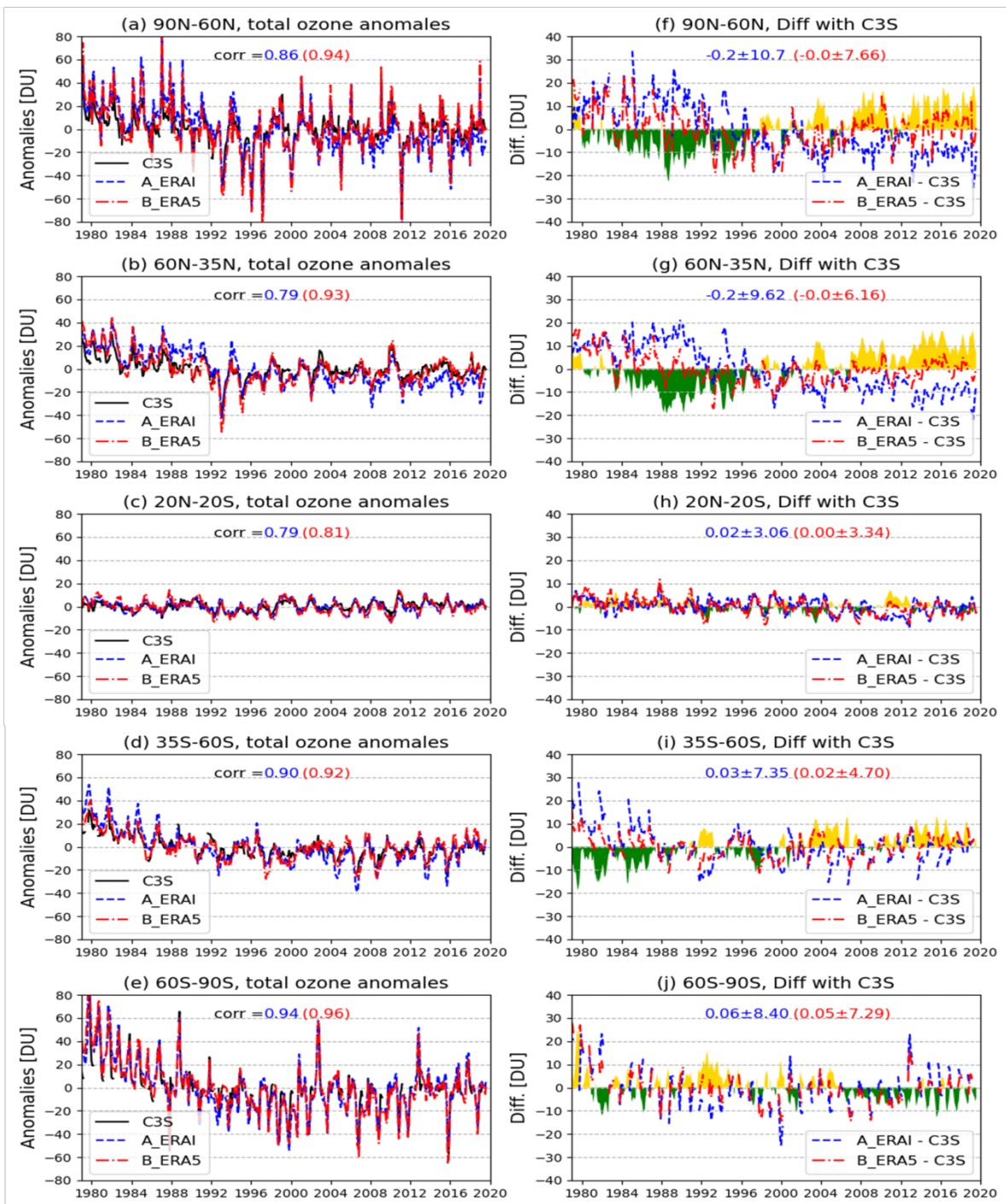

**Figure 1: (Left panels a-e)** Monthly mean total column ozone (TCO) anomalies (DU) derived from C3S (black solid line) and TOMCAT simulations A_ERAI (blue dashed line) and B_ERA5 (red dash-dot line) over 1979-2019 (August) for five latitude regions: 90°N-60°N, 60°N-35°N, 20°N-20°S, 35°S-60°S and 60°S-90°S. **(Right panels f-j)** Absolute differences in TCO between each simulation and C3S (blue dashed line for A_ERAI - C3S and red for B_ERA5 -

**C3S) as well as between the two simulations (B_ERA5 - A_ERAI, shaded with green colour for B_ERA5 < A_ERAI and yellow for B_ERA5 > A_ERAI). Correlation coefficients and TCO differences with standard deviations between simulation A_ERAI (B_ERA5) and C3S are shown with blue (red) text.**

### 3.1.2 Seasonal variability in total column ozone

**Figure 2** compares the C3S TCO with A_ERAI and B_ERA5 simulations over the period 1979-2018 to examine the
250 climatological seasonal cycle characteristics of TCO. As expected, both model simulations reproduce the major seasonal characteristics of the zonal mean distribution of C3S TCO (**Figures 2a-c**). Differences between the model simulations and C3S (**Figures 2d-e**) show that TCO in the tropics (especially north of the Equator) is underestimated in both simulations compared to C3S. Compared to the large negative biases (up to ~30 DU) seen in A_ERAI, TCO from B_ERA5 exhibits relatively smaller negative biases (< ~20 DU) in the tropics. In NH mid-high latitudes, A_ERAI overestimates the observed
C3S TCO across all seasons, while B_ERA5 shows somewhat larger positive biases (~15 DU) during NH winter-spring seasons but negligible biases during summer-autumn seasons. The comparison in **Figure 2f** shows that B_ERA5 exhibits positive TCO differences at mid-high latitudes during winter-spring seasons in both the hemispheres. This characteristic points to potential differences in the representation of tropics-to-mid-high-latitude ozone transport via the meridional circulation (the Brewer-Dobson circulation (BDC)) between the two reanalysis data sets. For example, positive differences
in **Figure 2f** during NH winter-spring seasons, and negative differences during summer-autumn seasons, indicate that on average wintertime ozone build-up and summertime ozone losses are significantly different between two model simulations. Also, during SH spring (September-October-November) slightly larger TCO in the tropics and smaller values at mid-latitudes in B_ERA5 indicate weaker ozone transport in ERA5. At the same time, larger TCO values in the SH polar cap during JJA (June-July-August) may indicate more mixing near the edge of the Antarctic polar vortex. The differences
between B_ERA5 and A_ERAI simulated TCO in the tropics and mid-high latitudes suggest that transport pathways between two reanalysis systems are different from the expected behaviour such as increased transport decreasing tropical ozone and increasing mid-high latitude ozone (e.g. Weber et al., 2003; Dhomse et al., 2006; Chrysanthou et al., 2019).

Next, we compare the stratospheric column ozone (SCO, integrated from 316 hPa to 1 hPa) over the period 1984-2018, which is shown in the supplementary **Figure S1**. We find that both simulations can reproduce the seasonal characteristics of
270 SCO from SWOOSH although they are overestimated for most mid-high latitude regions. The comparison of SCO between the two simulations shows consistent results with those for TCO in **Figure 2f**.

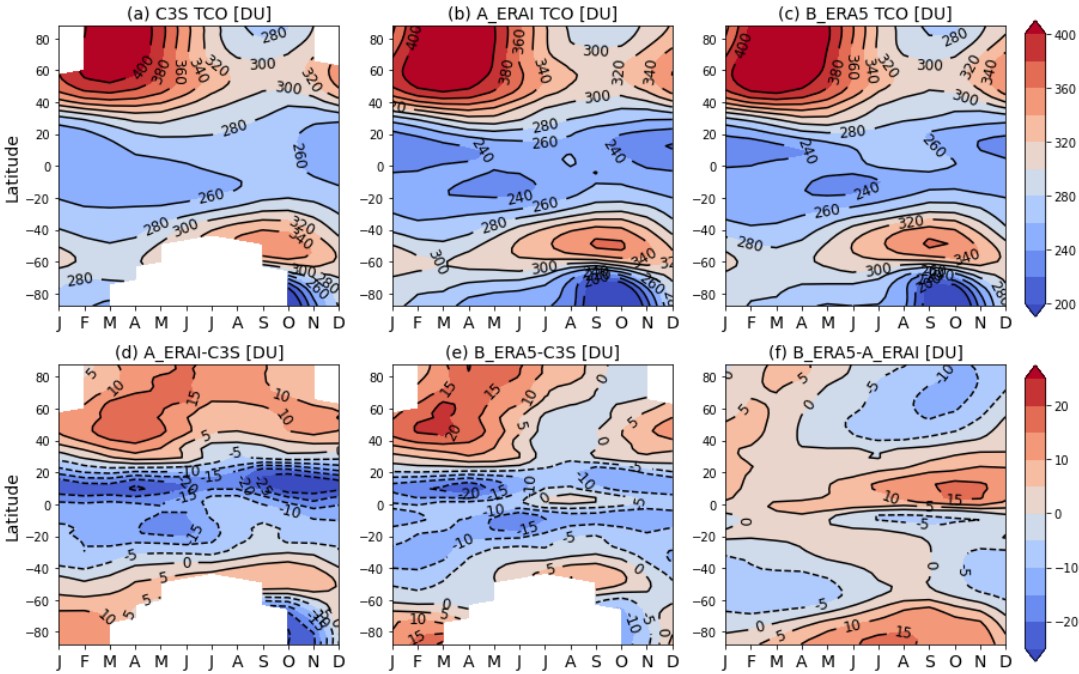

**Figure 2: Zonal and monthly mean TCO (DU) climatology over the period 1979-2018 based on (a) C3S and two model simulations (b) A_ERAI and (c) B_ERA5. The absolute differences between each simulation and C3S, as well as between the two simulations, are shown in (d) A_ERAI - C3S, (e) B_ERA5 - C3S and (f) B_ERA5 - A_ERAI, respectively.**

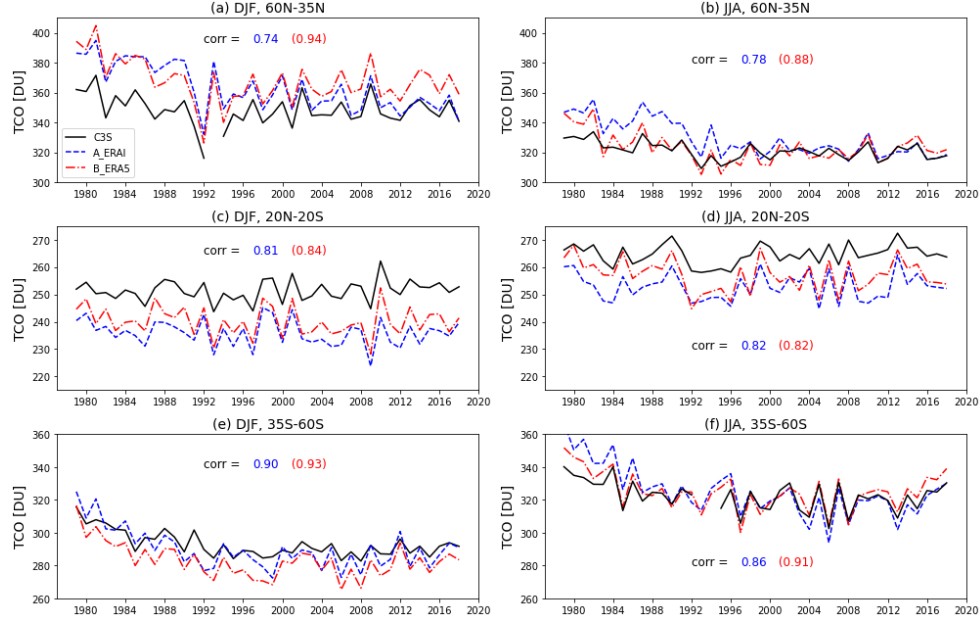

**Figure 3: December-January-February (DJF) and June-July-August (JJA) mean TCO (DU) for the period 1979-2018 from C3S (black solid line), A_ERAI (blue dashed line), and B_ERA5 (red dash-dot line) averaged over the latitude**

bands (a, b) 60°N-35°N, (c, d) 20°N-20°S and (e, f) 35°S-60°S. Correlation coefficients between simulation A_ERAI (B_ERA5) and C3S are shown in each panel with blue (red) text. Note that y-axes change for different panels.

**Figure 3** compares the TCO evolution during DJF and JJA seasons averaged over 60°N-35°N, 20°N-20°S and 35°S-60°S from C3S, A_ERAI and B_ERA5, and their differences are shown in the supplementary **Figure S2**. TCO variations from both CTM simulations show reasonable agreement with C3S data, in line with the results in **Figure 2**. As stratospheric transport is dominant in winter, there is a steady build-up in mid-high latitude TCO in both the hemispheres, whereas in summer, there is a steady decline due to photochemical loss (e.g. Fioletov and Shepherd, 2003; Tegtmeier et al., 2008). As noted earlier, both model simulations A_ERAI and B_ERA5 underestimate the observed DJF and JJA mean total ozone variability in the tropics, indicating either or both models have weaker ozone production and/or stronger ozone transport to mid-high latitudes. Differences between the two simulations (B_ERA5 - A_ERAI) remain within +10 DU in both DJF and JJA timeseries (**Figure S2 c-d**).

Focusing on the mid-latitudes (**Figures 2a-2b and 2e-2f**), the TCO from A_ERAI is more comparable with C3S in the SH mid-latitude band (with biases of -2.76±5.82 DU in DJF and 3.09±8.71 DU in JJA, respectively) but is overestimated in the NH mid-latitudes (14.39±10.30 DU in DJF and 7.99±8.43 DU in JJA), especially in the period until 1992. Additionally, B_ERA5 also overestimates DJF mean TCO in the NH mid-latitudes (19.07±6.14 DU biases), but shows negative biases in the SH mid-latitudes (-9.85±4.04 DU). Both models overestimate DJF mean TCO in the NH mid-latitudes, which might be due to the greater wintertime transport estimated in model simulations, as the poleward transport of ozone is most effective in the winter hemisphere (e.g. Chipperfield and Jones, 1999). B_ERA5 overestimates the observed DJF mean TCO in the NH mid-latitudes while it underestimates it in the SH mid-latitudes, which might be due to the differences in simulated wintertime ozone build-up (transport dominates) and summertime ozone losses (photochemical loss dominates) compared to observations. In JJA, B_ERA5 agrees better with C3S in both hemisphere mid-latitudes, except for the overestimation in the beginning and end years. Despite larger biases in B_ERA5 than in A_ERAI (e.g. **Figures 3a** and **3e**), the correlation coefficients between B_ERA5 and C3S are higher than A_ERAI, which suggests that A_ERAI might have some unrealistic annual variability. Consistent with the results of correlation analysis shown in **Figure 1,** which is based on monthly mean TCO anomalies, both simulations A_ERAI and B_ERA5 are better correlated with C3S in the SH than in the tropical and NH mid-latitude bands. Overall simulation B_ERA5 shows relatively better correlation with C3S in both seasons for all latitude bands.

### 3.1.3 Total column ozone trends and explanatory variables

To gain better insight about the implications to the ozone trend estimation due to differences discussed above, we apply the ILT-based multi-variate linear regression model (Section 2.3) to the DJF and JJA mean TCO time series to determine the long-term (1979-2018) ozone trends and changes over 60°N-35°N, 20°N-20°S and 35°S-60°S. The percentage ozone changes derived from peak contributions of different proxies in percentage ($(max.-min.)/mean \times 100$) are shown in **Figure 4**. Error bars indicate the confidence bounds at the 95% statistical significance level quantified by ± 2 standard

deviations (σ), and the negative and positive patterns come from the regression coefficients. As expected, the regression models for C3S and CTM simulations show negative trends at all latitude bands considered here before 1998 (Trend1), with

more significant decreases at NH and SH mid-latitude bands than C3S. The overestimation of these negative trends in model simulations could be due to (1) the unrealistic trends in stratospheric transport, especially during 1980s and 1990s, (2) the incomplete presentation of complex atmospheric processes and their feedbacks, (3) the incorrect parameterization for photochemical reactions in a CTM or (4) the uncertainties in observational data sets (e.g. Dhomse et al., 2021).

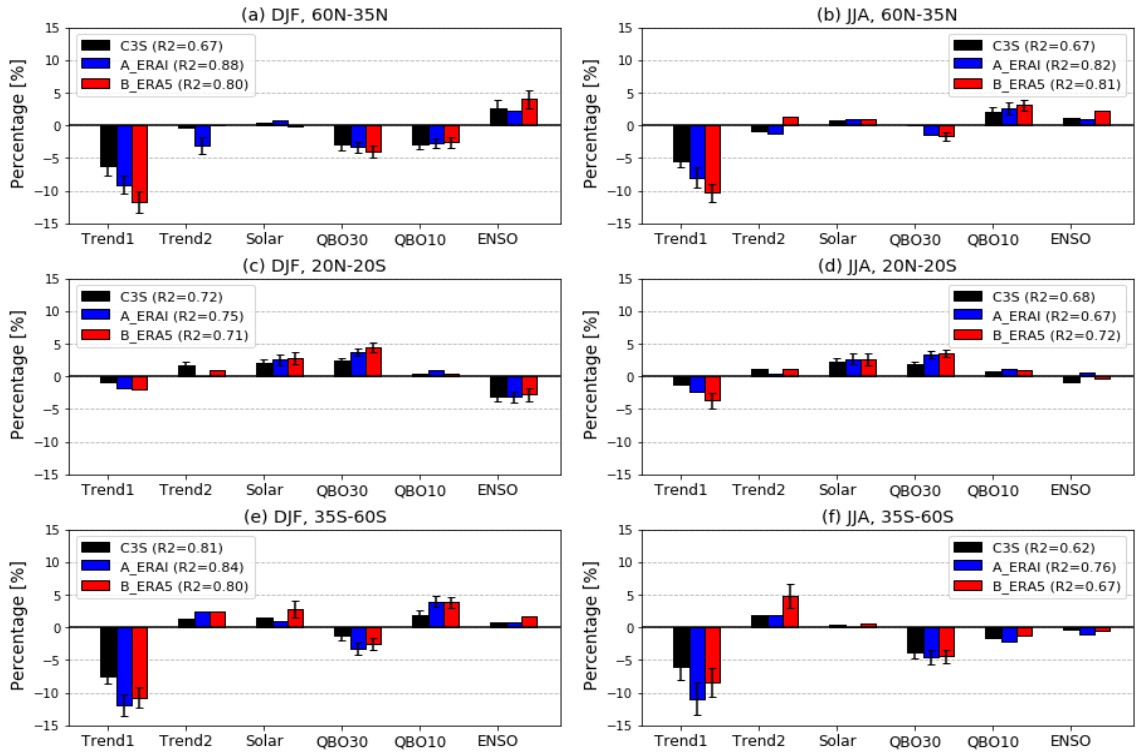

**Figure 4: Peak contributions (in %) from piecewise linear trend and explanatory variable terms (see equation (1)) to the total ozone column variability during DJF and JJA for (a, b) 60°N-35°N, (c, d) 20°N-20°S and (e, f) 35°S-60°S for C3S, A_ERAI and B_ERA5 during 1979-2018. Error bars indicate the confidence bounds at the 95% statistical significance level quantified by ± 2 standard deviations (σ). The determination coefficients (R-squared) of the regression**
**model for DJF and JJA mean TCO time series from C3S, A_ERAI and B_ERA5 over the 60°N-35°N, 20°N-20°S and 35°S-60°S regions are presented in each plot.**

The recovery since 1998 (Trend2) from C3S and both simulations differs from each other in terms of its magnitude and significance. C3S shows weak recovery for the tropical and SH mid-latitude bands except for the NH mid-latitude, with a significant recovery trend in DJF for the tropical region. Simulation A_ERAI also shows negligible and positive trends in the
tropical and SH mid-latitude regions, but the trends are negative at NH mid-latitudes with significant decrease in DJF. In contrast, B_ERA5 shows positive trends for all three latitude bands that are larger than 2-σ variance in the SH mid-latitudes

(JJA). These differences in ozone recovery can be linked to the differences between ERA5 and ERA-Interim forcing fields (such as trends in stratospheric transport processes) used in model simulations. It is also important to note that TOMCAT setup used here has simplistic representation of the tropospheric chemistry, so any long-term changes in the tropospheric column ozone (about 10% of TCO) may not be represented in both model simulations.

The differences in the proxy contributions for the DJF and JJA seasons are consistent with our understanding that total ozone variability is dominated by different processes in winter and summer. We also find slight differences in proxy contributions to the total ozone variability from C3S, A_ERAI and B_ERA5, but to a large extent contributions from the solar cycle, QBO and ENSO in the ozone variability are somewhat similar. For example, positive QBO anomalies in the tropics and negative anomalies in the subtropical regions are associated with the QBO phase change from the equator to the subtropics (e.g. Chehade et al., 2014).

## 3.2 Variability and trends in ozone profiles

### 3.2.1 Comparison of vertical ozone profiles

We now compare ozone profiles from model simulations and SWOOSH dataset. **Figure 5** shows vertical profiles of ozone averaged over 60°N-35°N, 20°N-20°S and 35°S-60°S latitude bins, along with the relative differences for each model simulation with respect to SWOOSH for the whole time period (1984-2018) as well as for DJF and JJA seasons. In all cases, both A_ERAI and B_ERA5 underestimate upper stratospheric ozone concentrations (10-1 hPa), while overestimate the lower stratospheric ozone concentrations (147-32 hPa for the mid-latitudes and 100-32 hPa for the tropics) to varying degrees. These biases might be associated with deficiencies in the representation of the photochemical reactions and dynamical processes in the model (e.g. Mitchell et al., 2020; Dhomse et al., 2013; 2016; 2021).

Overall, simulation B_ERA5 shows larger negative biases in the upper stratosphere (up to ~ -10% at 3 hPa) than does A_ERAI. In the middle stratosphere (32-10 hPa), both simulations are in good agreement with each other. The biases between model simulations and SWOOSH in the lower change with latitude bands and seasons. In the tropical lower stratosphere (~80 hPa), B_ERA5 shows larger (> 40%) biases than those in A_ERAI (~ 20%) for both DJF and JJA seasons. Although B_ERA5 shows better correlation with the observed tropical TCO and smaller differences than A_ERAI does, the comparison in tropical ozone profiles indicates that w. r. t. SWOOSH, B_ERA5 has larger biases in both the upper and lower stratosphere. In the NH mid-latitude lower stratosphere (~100 hPa), B_ERA5 exhibits slightly more positive biases w.r.t. SWOOSH in DJF (boreal winter) but smaller biases in JJA than A_ERAI does. In the SH mid-latitude lower stratosphere, A_ERAI shows larger biases in DJF (austral summer) but the biases in JJA for both simulations are comparable. The comparison of ozone changes between two simulations indicates that their differences in the lower stratosphere largely contribute to TCO differences shown in **Figures 1** and **2**. In the lower stratosphere ozone is long-lived and under dynamical control, indicating the effects of changes in background meteorological forcing fields on simulated lower stratospheric ozone.

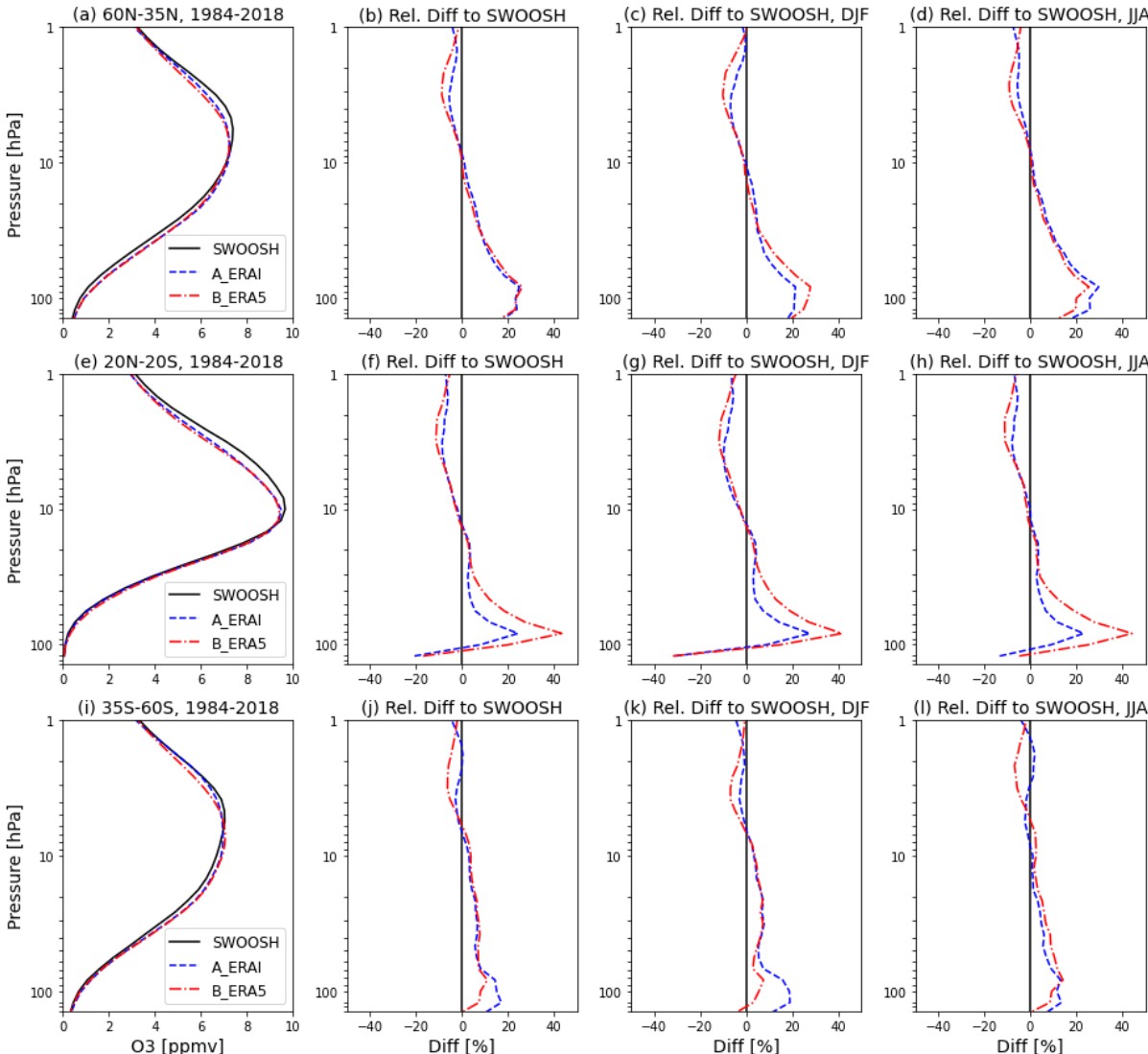

**Figure 5: Averaged vertical ozone profiles from SWOOSH (black solid line), A_ERAI (blue dashed line) and B_ERA5 (red dash-dot line) for (a-d) 60°N-35°N, (e-h) 20°N-20°S, and (i-l) 35°S-60°S (1984-2018). Relative differences (%) referencing each simulation to SWOOSH averaged in the whole time period as well as DJF and JJA seasons are shown in the three right-hand columns for comparison.**

**3.2.2 Interannual variability of ozone and temperature anomalies**

After analysing biases in mean ozone profiles, we diagnose the time-dependent differences in ozone anomalies between two simulations and SWOOSH data set as shown in **Figure 6.** The relative differences between simulated and observed ozone anomalies are calculated with respect to SWOOSH ozone values. The comparison shows that simulation A_ERAI significantly overestimates the observed NH mid-latitude ozone anomalies for the early years (1984-1996) over the whole

stratosphere especially in the lowermost stratosphere. Afterwards ozone anomalies in A_ERAI are underestimated, while

ozone anomalies in B_ERA5 are more comparable with the observations except for the significant overestimation in the lower stratosphere during the later period 2006-2019 (August). The situation in the SH mid-latitude region is similar to that in the NH except that the biases are relatively smaller. These results are consistent with the comparison of TCO anomalies shown in **Figures 1g** and **1i**, also indicating that differences in the lower stratosphere are mainly responsible for their

differences in TCO. In the tropics, both simulations underestimate the observed ozone anomalies in the lower stratosphere before 2000 but overestimate them afterwards. However, in the upper stratosphere (above 3 hPa) cases are opposite for the two simulations, which might be associated with the uncertainties in temperature-dependent reaction rates in the models (e.g. Stolarski et al., 2010; Dhomse et al., 2013, 2016). We also compare the profiles of correlation coefficients between the simulated and SWOOSH ozone anomalies over the latitude bands 60°N-35°N, 20°N-20°S and 35°S-60°S, as shown in

**Figure S3**. Again, though simulation B_ERA5 generally shows better correlation with C3S TCO anomalies (**Figure 1**), it does not show better correlation for the stratospheric ozone profile anomalies overall (e.g. in the upper stratosphere for the tropics and NH mid-latitudes and in the lower stratosphere for the SH mid-latitudes).

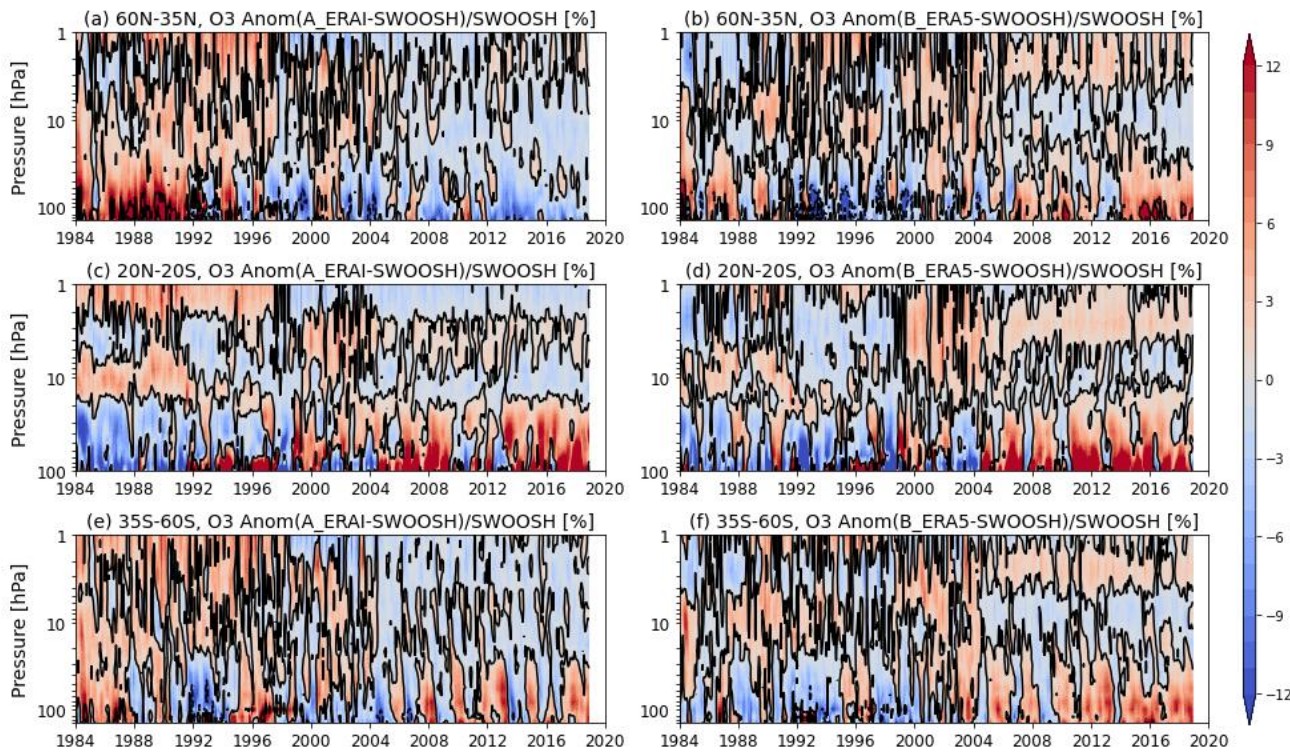

**Figure 6: Pressure-time cross section of the relative differences (%) in ozone anomalies between model simulations A_ERAI, B_ERA5 and SWOOSH over 1984-2019 (August) for different latitude regions (a, b) 60°N-35°N, (c, d) 20°N-20°S and (e, f) 35°S-60°S.**

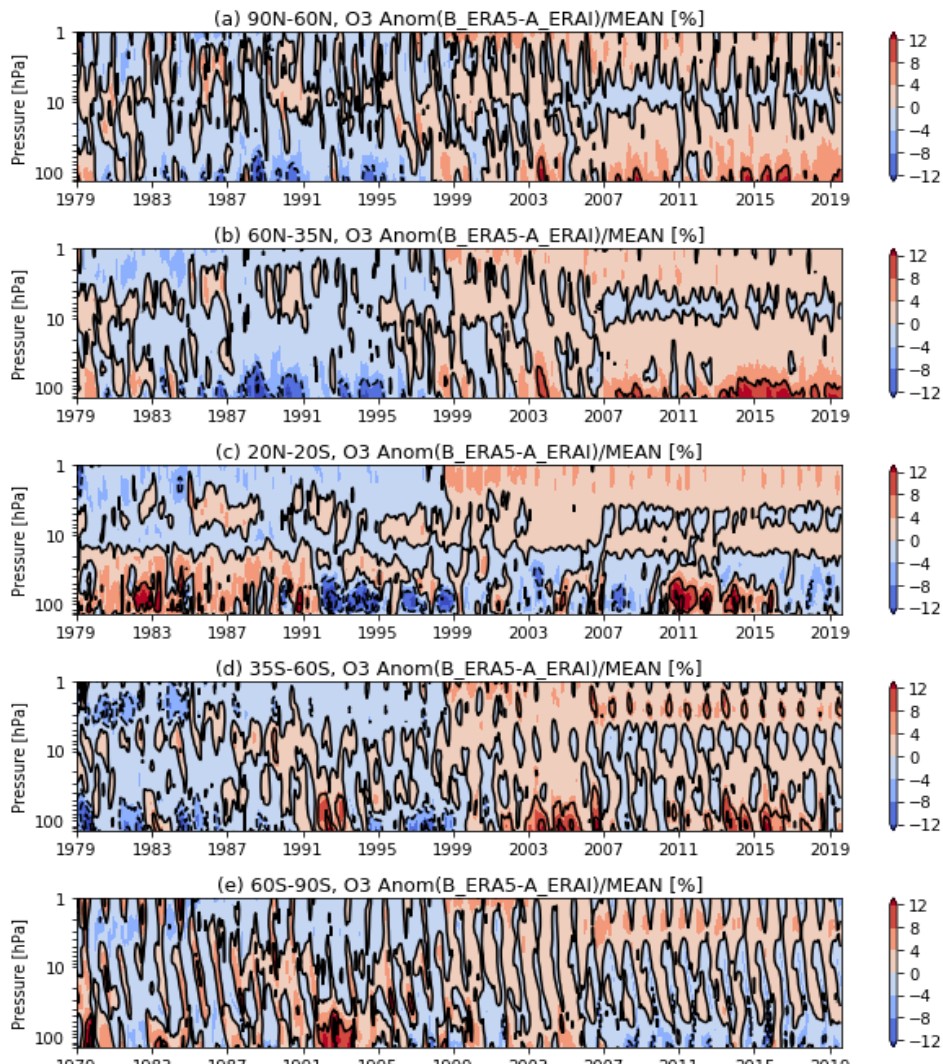

**Figure 7: Pressure-time evolution of the relative differences (%) in ozone anomalies between A_ERAI and B_ERA5 over 1979-2019 (August) for different latitude regions (a) 90°N-60°N, (b) 60°N-35°N, (c) 20°N-20°S, (d) 35°S-60°S and (e) 60°S-90°S.**

**Figure 7** shows the relative differences between A_ERAI and B_ERA5 for five latitude bands from 147 hPa to 1 hPa over the time period 1979-2019 (August). The positive differences in the upper stratosphere after 1998 for all latitude regions can clearly be seen, which means that upper stratospheric anomalies in simulation B_ERA5 are overestimated compared to A_ERAI. In the NH mid-high latitudes, the relative differences in the lower stratospheric ozone between the two simulations (B_ERA5-A_ERAI) also change from negative before 1998 to positive afterwards. These differences in the NH stratosphere (when integrated) are consistent with the characteristics seen in TCO anomalies as shown in **Figures 1f-g**. In the tropical lower stratosphere, B_ERA5 overestimates the ozone anomalies during the periods 1979-1991 and 2010-2016, and

405 underestimates in other periods. The situation in the SH mid-latitude lower stratosphere is similar to that in the NH mid-latitude where the biases between two simulations change from negative to positive around 2000, while it is not the case in the SH polar region.

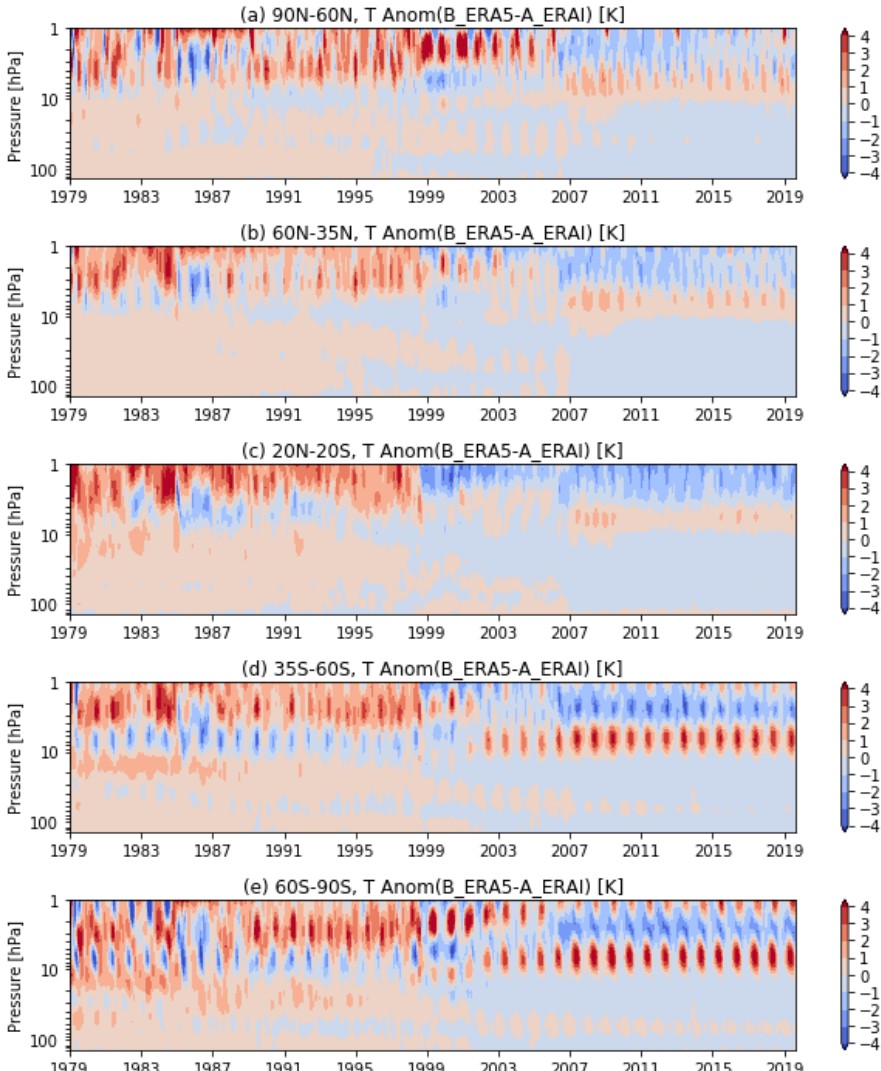

 **Figure 8: Similar to Figure 7 but for differences in temperature anomalies (K) between A_ERAI and B_ERA5 (B_ERA5- A_ERAI). Note the simulation B_ERA5 uses ERA5.1 reanalysis for the period 2000-2006.**

As ozone loss reactions are temperature dependent (e.g. Randel and Cobb, 1994; Douglass et al., 2012), in **Figure 8** we compare the temperature anomalies between A_ERAI and B_ERA5 to account for the relative differences in ozone anomalies in a similar fashion to **Figure 7**. Large biases in temperature anomalies between two simulations (B_ERA5- 415 A_ERAI) appear in the upper stratosphere for all latitude regions until around 1998, confirming some of the inhomogeneities

seen in ERA-Interim upper stratospheric temperatures (Dhomse et al., 2011; McLandress et al., 2014). Some recent studies argue that there has been significant improvements in ERA5 temperatures as it includes more measurements and uses updated bias correction techniques, model physics and CMIP5 radiative forcings in a 4Dvar data assimilation system (Hersbach et al., 2020; Simmons et al., 2020; Marlton et al., 2021). Besides, ERA5 has a higher top layer up to ~80 km with finer vertical resolution in the upper stratosphere than ERA-Interim which only extends up to ~65 km. The update in the radiation scheme and the improvement in the wind extrapolation scheme in ERA5 also mitigates erroneous temperatures compared to ERA-Interim (Hersbach, 2020 and references therein). Thus, the differences in the upper stratospheric temperatures from the reanalysis data sets drive the differences in ozone anomalies in this region, as cooler (warmer) temperatures causes more (less) ozone when photochemical processes dominate (e.g. Stolarski et al., 2012). In the lower stratosphere, however, temperature differences between the two simulations are relatively small and similar for all latitude bands, which cannot explain the differences in the lower stratospheric ozone anomalies. This corroborates the fact the ozone variability in the lower stratosphere depends on a more complex dynamical and photochemical processes and associated feedback pathways than that in the upper stratosphere.

### 3.2.3 Comparison of ozone profile trends

**Figure 9** shows the two independent linear trends for the zonal mean ozone anomalies over the periods 1984-1997 (Trend1) and 1998-2018 (Trend2) obtained from SWOOSH, A_ERAI and B_ERA5 simulations. Both A_ERAI and B_ERA5 reproduce the decreasing ozone trends before 1998, with some exceptions such as the inconsistent positive trends in the tropical region and the overestimated decline in the extratropical lower stratosphere relative to SWOOSH. The significant inconsistencies in the tropical region suggest that both model simulations are unable to reproduce SWOOSH type variations in the tropical lower stratosphere. It is also important to note that much smaller ozone concentrations in this region means larger retrieval errors for satellite measurements that are used in SWOOSH data set. For example, Davis et al. (2016) found that below 100 hPa, HALOE and SAGE III data sets show up to -60% and +20% biases w.r.t. collocated ozonesonde measurements in the tropics. Both the simulations also overestimate the downward trend in the extratropical lower stratosphere that partly explains the overestimated decline in simulated TCO (Trend1) in the NH and SH mid-latitude regions (**Figure 4**).

For the later period (1998-2018), both simulations show the increasing trends in the upper stratosphere that are consistent with SWOOSH-derived trends. Harris et al. (2015) argued that this increase is associated with stratospheric cooling and an almost linear decrease in stratospheric chlorine loading. In the lowermost stratosphere near 100 hPa, both SWOOSH and A_ERAI show negative trends (SWOOSH being more negative) in the tropical and NH extratropical regions, while B_ERA5 shows weak recovery or negligible trends. Large disagreement between SWOOSH and A_ERAI appears near 40 hPa where A_ERAI shows significant recovery opposite to the decrease in SWOOSH. However, in the SH mid-latitude lower stratosphere, both simulations show opposite recovery trends to the decrease in SWOOSH. Furthermore, the much stronger recovery signal in the Antarctic lower stratosphere in A_ERAI suggests that the agreement of simulation A_ERAI with

SWOOSH is hemispherically asymmetric. Similar to the increasing mid-latitude trends found in most CCMs (Ball et al.,
2020; Dietmüller et al., 2021), the increasing NH mid-latitude trends in simulation B_ERA5 indicate possible discrepancies
in ERA5 dynamical fields especially in the lower stratosphere.

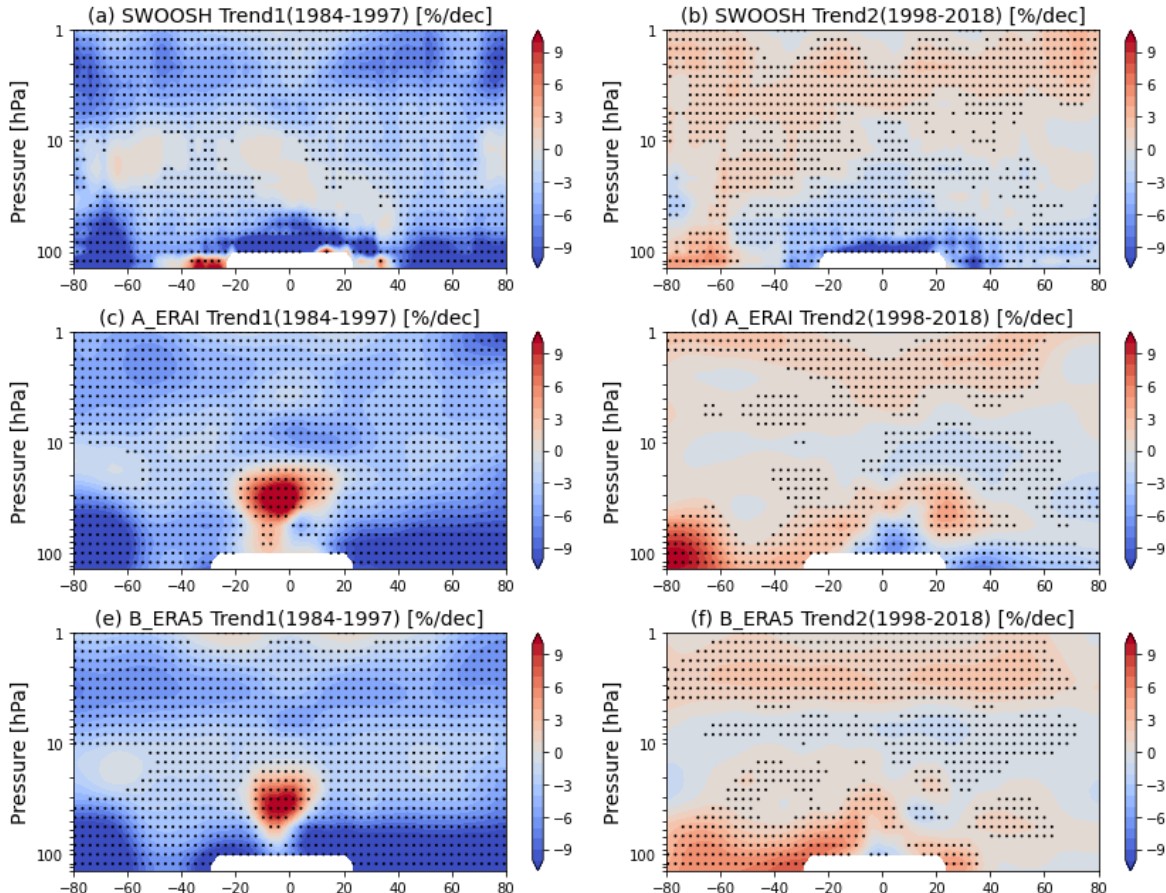

**Figure 9: Latitude-pressure cross sections of the piecewise linear trends of ozone anomalies (%/decade) over the**
**periods 1984-1997 and 1998-2018 for (a, b) SWOOSH, (c, d) A_ERAI and (e, f) B_ERA5, respectively. Stippled**
**regions indicate where the trends are statistically significant at 95% level of confidence.**

Zonally averaged linear trends for 60°N-35°N, 20°N-20°S and 35°S-60°S from SWOOSH, A_ERAI and B_ERA5 are
shown in the supplementary **Figure S4** as they quantitatively illustrate the long-term changes over two different periods:
1984-1997 (Trend1) and 1998-2018 (Trend2). During 1984-1997, SWOOSH ozone data show a consistent decrease in the
whole stratosphere across all three latitude bands considered here. Simulations A_ERAI and B_ERA5 are able to reproduce
negative ozone trends, especially in the SH mid-latitude stratosphere. However, both simulations overestimate the decline in
the NH mid-latitude lower stratosphere (with trends of -15±1.9% per decade at 100 hPa), but trends are underestimated in
the upper stratosphere (above 5 hPa). Surprisingly, model simulations even show opposite increasing ozone in the tropical

low-middle stratosphere between 15 and 50 hPa. During 1998-2018 almost all individual data sets show positive ozone trends in the upper stratosphere (1-5 hPa), with the largest recovery trend (~2.5% per decade) at 1 hPa from A_ERAI and at 2~3 hPa from B_ERA5. Again, larger discrepancies appear in the lower stratosphere at all latitudes. In contrast to the negative trends in the NH mid-latitude region seen in SWOOSH and A_ERAI, B_ERA5 shows small but positive trends. The positive trends that also appear in the tropical and SH mid-latitude regions for both model simulations are overestimated in B_ERA5. Hence, these results show that ozone trends from both simulations A_ERAI and B_ERA5 should be considered with care.

## 4 Mean age-of-air

As air parcels exhibit long residence times in the stratosphere, stratospheric mean age-of-air (AoA) provides a useful insight into the stratospheric transport processes. In a model, it is simulated simply by releasing an inert tracer from the tropical tropopause (e.g. Hall et al., 1999; Monge-Sanz et al., 2013, 2022). Simulated AoA are evaluated against observations and is considered as a standard test for stratospheric models (Waugh and Hall, 2002). Changes in AoA in the stratosphere mirror changes associated with the stratospheric mean meridional circulation (Stiller et al., 2008; Mahieu et al., 2014; Prignon et al., 2021). It should be noted that AoA captures the combined effects of the advective part of the BDC known as the residual circulation and the two-way mass exchange (mixing) on stratospheric tracer transport (Plumb, 2002; Shepherd, 2007), the effects of which might counteract each other, especially in the lowermost stratosphere (Birner and Bönisch, 2011; Garney et al., 2014; Karpechko et al., 2018). The interannual and long-term changes in the strength of the BDC are responsible for the winter-spring build-up patterns in extratropical ozone (e.g. Fusco and Salby, 1999; Weber et al., 2003; Dhomse et al., 2006).

### 4.1 Comparison of mean age distributions

Ploeger et al. (2021) analysed the global stratospheric BDC using simulations of stratospheric mean AoA with the Chemical Lagrangian Model of the Stratosphere (CLaMS) driven by reanalysis (ERA5/ERA-Interim) winds and total diabatic heating rates. They found that ERA5-based results exhibit older AoA compared to results based on ERA-Interim, indicative of a significantly slower BDC for ERA5. Prignon et al. (2021) investigated the BDC variability and long-term changes using inorganic fluorine simulated by the Belgian Assimilation System for Chemical ObsErvation chemistry transport model (BASCOE CTM) driven by 5 modern reanalyses. The comparison with observations suggests an overall better representation of transport variability in ERA5 than in ERA-Interim over the period 1990-2018, especially in the NH mid-latitudes. As discussed earlier in our ozone trend analysis (Section 3.2.3), we find B_ERA5 shows a significant increasing trend in lower stratospheric ozone at NH mid-latitudes, while observations (SWOOSH) and A_ERAI continue to decrease after 1998. Hence, we diagnose the effect of changes in the representation of stratospheric transport by analysing variability and trends in the AoA tracer between two simulations and explore the potential causes for these inconsistencies.

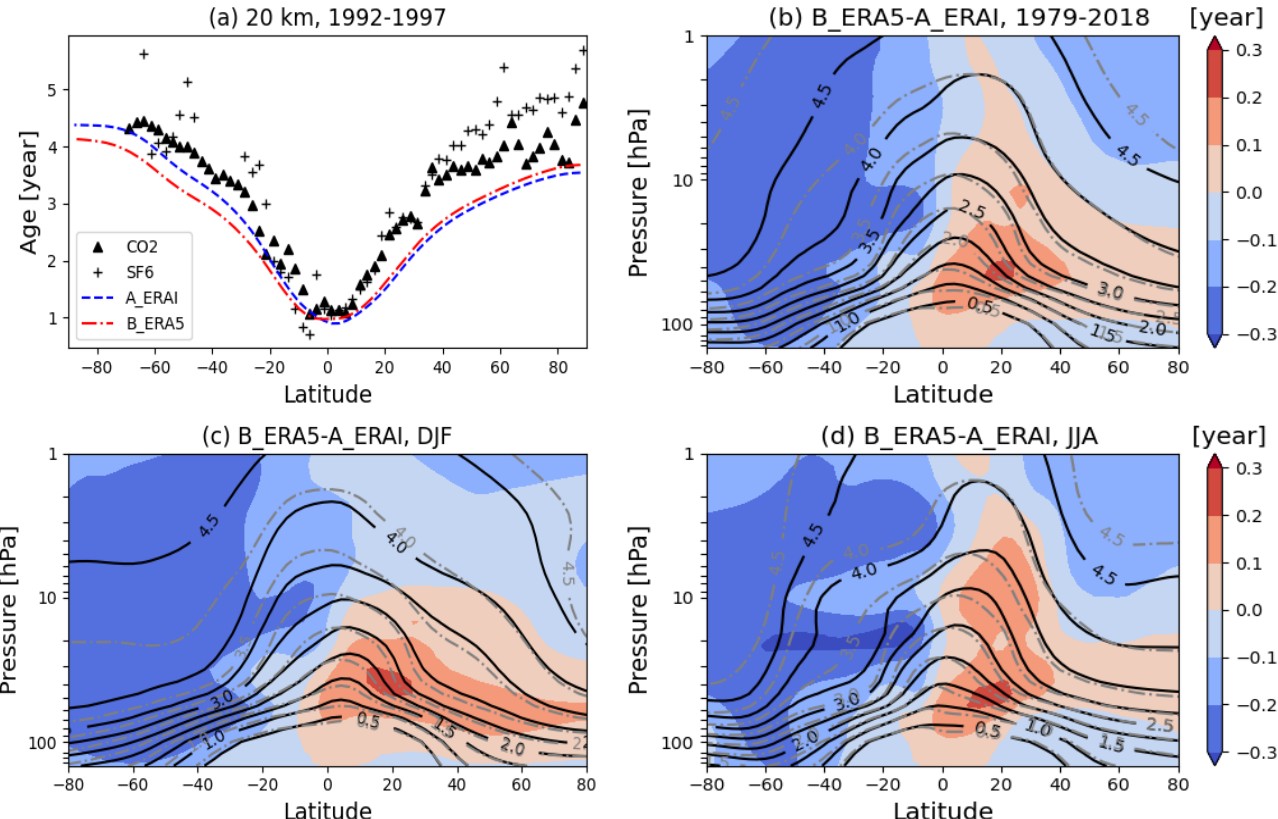

**Figure 10: (a)** Mean age-of-air (AoA, years) at 20 km at different latitudes from in situ observations of $CO_2$, $SF_6$ (black symbols, from Hall et al., 1999), A_ERAI (blue solid line) and B_ERA5 (red dash-dot line). **(b)** Pressure-latitude cross section of mean age from A_ERAI (black solid contours) and B_ERA5 (grey dash-dot contours), and their differences (B_ERA5 - A_ERAI, in red and blue shading) averaged over 1984-2018. Panels (c) and (d) are similar to (b) but for DJF and JJA means, respectively.

**Figure 10a** shows mean AoA at 20 km from model simulations as well as in situ $CO_2$ and $SF_6$ measurements described in Hall et al. (1999). The mean AoA from A_ERAI and B_ERA5 simulations over the period 1992-1997 agree relatively well with the in situ data (better with $CO_2$), and both simulations show steeper gradients in AoA at SH mid-latitudes relative to NH mid-latitudes. We find that both simulations underestimate the observed mean age, especially at NH mid-latitudes. As shown in Chipperfield (2006), the use of potential temperature ($\theta$) coordinates in the stratosphere can improve low-biased stratospheric AoA in the model using hybrid sigma-pressure ($\sigma$-p) levels. The general characteristics of the stratospheric mean age (**Figure 10b**) are evident for both A_ERAI and B_ERA5 simulations, with age increasing with both latitude and altitude (Ploeger et al., 2019, 2021). The comparison of the mean age shows that age from B_ERA5 is slightly older than that from A_ERAI in the NH stratosphere but somewhat younger in the SH stratosphere, which suggests a slower BDC in the NH but a faster BDC in the SH.

The integrated effect of BDC transport in A_ERAI and B_ERA5 is compared for mean AoA between winter and summer seasons in **Figures 10c-d**. The DJF and JJA mean comparisons are consistent with **Figure 10b**. However, in DJF (boreal wintertime), B_ERA5 shows slightly older air than A_ERAI (~0.14 year at 20 km) when compared to boreal summertime (~0.01 year at 20 km). This contrasting feature indicates some fundamental differences in the representation of BDC between two reanalysis data sets. It also highlights that a slower BDC might not reduce wintertime ozone build-up at NH mid-high latitudes and B_ERA5 also shows improvements in the TCO biases in the tropics. A possible explanation is that the finer vertical resolution in ERA5 significantly alters vertical transport pathways that are critical for controlling ozone concentration as within a few kilometres in the stratosphere the ozone lifetime changes from days to a few years.

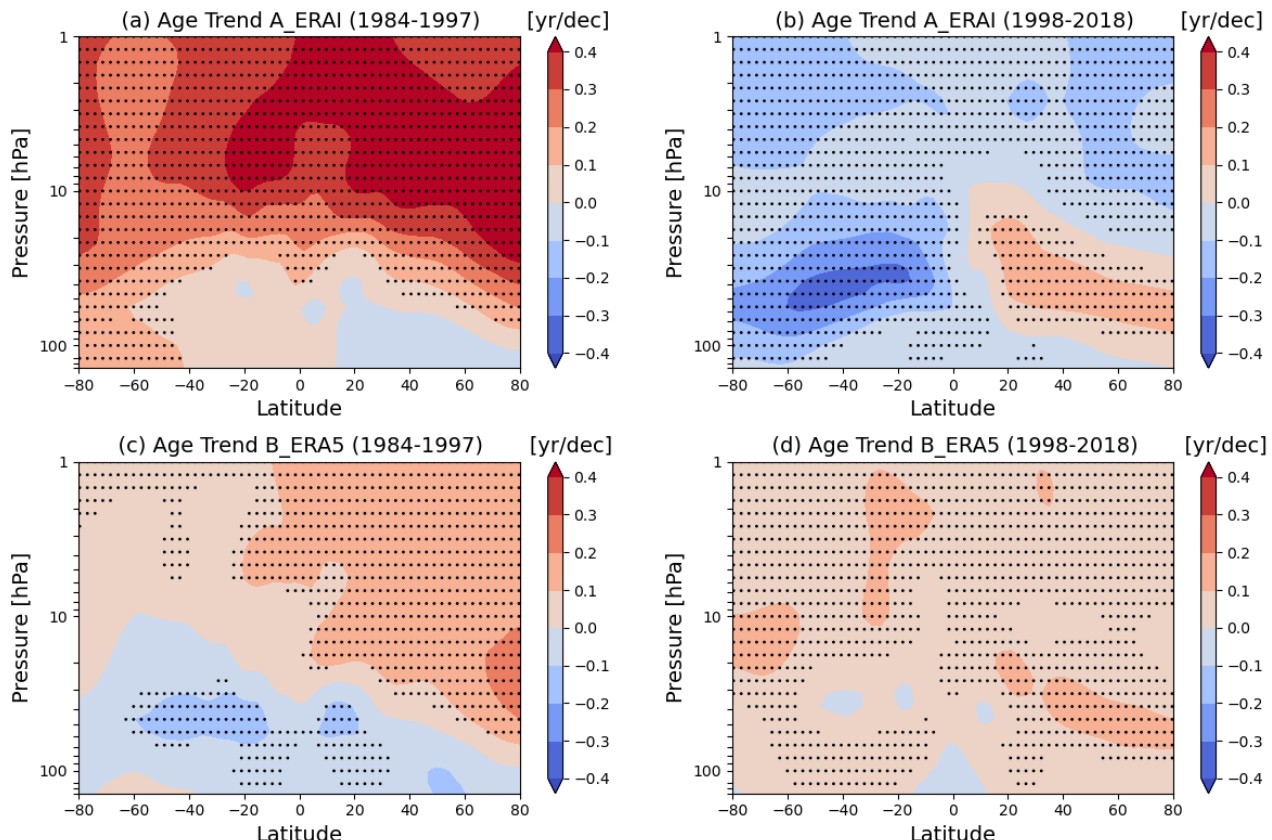

**Figure 11: Mean age-of-air trends (year/decade) for the period 1984-1997 from simulations (a) A_ERAI and (c) B_ERA5. Panels (b) and (d) are the same as (a) and (c), respectively, but for the period 1998-2018. Stippled regions indicate where the trends are statistically significant at 95% level of confidence.**

### 4.2 Comparison of mean age trends

AoA trends over the periods 1984-1997 and 1998-2018 from A_ERAI and B_ERA5 are shown in **Figure 11**, corresponding to the trends in ozone shown in **Figure 9**. Mean AoA trends are calculated using linear regression on the

deseasonalized time series. As shown in **Figures 11a** and **c**, both A_ERAI and B_ ERA5 simulations show increasing AoA over the 1984-1997 period in the upper and middle stratosphere especially in the NH (about 0.2-0.4 year/decade). A closer look at the differences suggest weaker positive AoA trends in the upper stratosphere and larger negative trends in the lower

stratosphere in B_ERA5 compared with A_ERAI. This can be confirmed by the differences of the two simulated deseasonalized AoA time series, as shown in **Figure 12**, with biases in B_ERA5 changing from positive to negative over 1984-1997.

During 1998-2018, A_ERAI shows clear positive trends in the NH and negative trends in the SH lower stratosphere (**Figure 11b**). The hemispheric dipole trend pattern seen in A_ERAI AoA are similar to the earlier studies (Haenel et al.,

2015; Stiller et al., 2017; Ploeger et al. 2021; Monge-Sanz et al., 2022). In contrast, B_ERA5 (**Figure 11d**) shows increasing AoA trends in the whole stratosphere, indicating a decelerating BDC. The globally positive AoA trends in B_ERA5 can also be seen from **Figure 12** in which B_ERA5 shows positive biases since 2012 compared to A_ERAI. It should be noted that the positive AoA trends seen in B_ERA5 throughout the stratosphere are opposite to the negative ERA5 trends (over the 1989-2018 period) shown in Ploeger et al. (2021). They suggested the clear decrease in ERA5 mean age is not a simple

linear trend and appears to be related to the increased AoA values at the beginning of the period and the step-like decreases during the 1990s. The remarkable differences in B_ERA5 mean AoA values and trend estimates (**Figures 10a** and **11d**) from the CLaMS model simulations in Ploeger et al. (2021) might be due to the different horizontal resolutions, p or θ coordinates, and/or calculation methods used. However, the differences in mid-latitude AoA trends from A_ERAI and B_ERA5 over the 1998-2018 period appear more consistent with the inorganic fluorine trends based on BASCOE CTM simulations for the

2004-2018 period (Prignon et al., 2021).

The increasing AoA in B_ERA5 after 1998 as well as the older age in the NH lower stratosphere, suggest that other transport pathways (such as downward transport/reduced transport in the troposphere) might have been responsible for the increasing ozone in the NH extratropical lower stratosphere in B_ERA5. A possible explanation might be changes in the vertical resolution (and changes in number and type of observations) assimilated in ERA5. For example, in NH mid-high

latitude, B_ERA5 shows somewhat older AoA in the lowermost stratosphere (near 100 hPa), but between 70 to 10 hPa, B_ERA5 shows slightly younger AoA compared to A_ERAI. In contrast, B_ERA5 minus A_ERAI ozone differences seen in **Figure 7 (a, b)** remain positive throughout the stratosphere with an exception of slightly negative values near 10 hPa. This clearly shows that changes in vertical transport can lead to larger changes in lower stratospheric ozone as ozone lifetime increases exponentially from a few days near middle stratosphere to a few years in the lower stratosphere. A similar feature

is observed in the SH mid-latitudes. However, at SH high-latitudes, B_ERA5 shows somewhat positive AoA compared to A_ERAI but simulated ozone differences are negative in the lower-middle stratosphere. Additionally, **Figure 12** also shows that mean AoA anomalies in B_ERA5 show negative biases compared to A_ERAI from 1992 to around 2011, which is somewhat similar to the step-like changes in Ploeger et al. (2021). These changes might be associated with the representation of Mt. Pinatubo volcanic eruption induced circulation/chemistry changes (e.g. Poberaj et al., 2011; Dhomse et al., 2015;

Monge-Sanz et al., 2022), transport processes as well as changes in number of observations used between these two data assimilation systems (e.g. Fujiwara et al, 2017).

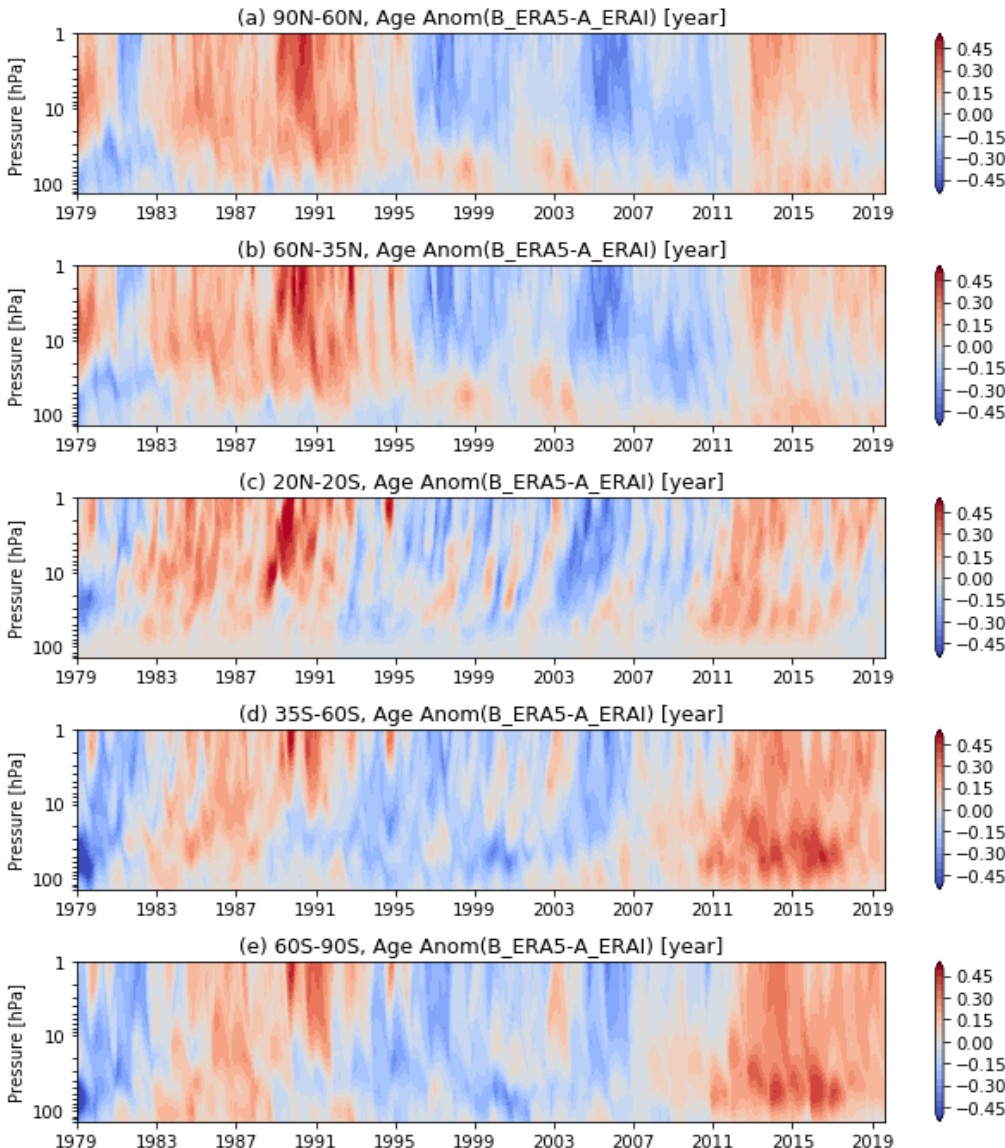

**Figure 12: Pressure-time series of differences in mean age-of-air (AoA) between A_ERAI and B_ERA5 (B_ERA5 - A_ERAI) over 1979-2019 (August) for (a) 90°N-60°N, (b) 60°N-35°N, (c) 20°N-20°S, (d) 35°S-60°S and (e) 60°S-90°S**
**zonal regions. Data have been deseasonalized by applying a 12-month running mean.**

## 5 Summary and conclusions

We have investigated the performance of two TOMCAT model simulations (A_ERAI and B_ERA5) forced with different ECMWF reanalysis datasets (ERA-Interim and ERA5). The variability and trends in total column ozone and stratospheric ozone profiles are compared with the observation-based datasets (C3S and SWOOSH). We also analysed an AoA tracer to diagnose the impact of stratospheric transport processes on simulated ozone. Our main results are summarized as follows:

- Comparison with C3S total column ozone anomalies (1979-2019) suggests that simulation B_ERA5 shows better agreement than A_ERAI. Largest biases between the A_ERAI and B_ERA5 model simulations appear in the NH mid-high latitudes. In the tropics (20°S-20°N), both simulations underestimate the observed TCO, and B_ERA5 shows some improvements compared to the larger negative biases seen in A_ERAI. During winter-spring seasons B_ERA5 shows larger positive biases in both the hemispheres, which suggests differences in representation of the stratospheric Brewer-Dobson circulation between these two reanalysis data sets. The ILT-based regression model shows that compared to C3S-based trend estimates, both A_ERAI and B_ERA5 overestimate the negative trends before 1998 at both hemispheric mid-latitude bands and B_ERA5 overestimates the recovery since 1998.

- Compared to SWOOSH vertical ozone profiles (1984-2019), both A_ERAI and B_ERA5 underestimate the observed upper stratospheric ozone concentrations while they overestimate the middle and lower stratospheric ozone to varying degrees. B_ERA5 shows larger ozone biases in the tropics in both the upper and lower stratosphere. The larger biases between simulations A_ERAI and B_ERA5 appear in the lower stratosphere, where ozone concentrations are primarily controlled by dynamical processes that largely translate into the biases seen in total column ozone. The differences in the upper stratospheric ozone anomalies between the two simulations are anti-correlated with the differences of temperature anomalies in the upper stratosphere, while ozone variability in the lower stratosphere is much more complicated. The ILT-based regression model shows that although simulation A_ERAI is consistent with SWOOSH with negative trends in the lowermost stratosphere in the tropical and NH mid-latitude regions since 1998, there still exist large differences with more significant positive trends seen in A_ERAI at ~40 hPa in the NH extra-tropics and in the Antarctic, while B_ERA5 shows inconsistent increasing trends in both NH and SH mid-latitude regions. Hence, trends derived using either simulation should be considered with care.

- Analysis of the AoA tracer suggests that both A_ERAI and B_ERA5 underestimate the observation-based mean age, at NH mid-latitudes. Simulation B_ERA5 shows somewhat older AoA in the NH stratosphere but younger in the SH stratosphere compared to A_ERAI. Older air in B_ERA5 in the NH lower stratosphere, especially during boreal winter (DJF), indicates a slower BDC. However, this does not translate in reduced wintertime ozone build-up suggesting key differences between horizontal as well as vertical transport pathways between these two reanalysis data sets. During 1998-2018, A_ERAI shows a hemispheric dipole trend pattern with increasing AoA in the NH and decreasing trend in the SH lower stratosphere. In contrast, B_ERA5 shows increasing AoA in the whole stratosphere. The increasing AoA

in B_ERA5 after 1998 and the older age in the NH lower stratosphere suggest other transport pathways might be responsible for the increasing ozone in the NH lower stratosphere.

Our results show that although simulation B_ERA5 shows better agreement with observed TCO anomalies compared to A_ERAI, there still exist larger biases over certain regions (e.g. NH in winter). Similarly, although simulation A_ERAI is more consistent with SWOOSH with negative trends in the lowermost stratosphere in the tropical and NH mid-latitude regions post-1998, there also exist larger biases over certain regions (e.g. ~40 hPa in the NH extra-tropics). With the newer reanalysis, B_ERA5 does not perform better in simulating stratospheric ozone overall, and both simulations should be

treated carefully for trend estimates. The association between the simulated ozone differences and age-of-air differences suggests that simulation B_ERA5 may not yet be capable to reproduce the trend and strength of the stratospheric circulation (BDC) changes.

*Data availability*. The satellite and climate data used in this study are available at the sources and references in the dataset

section. The model data used for the figures in this paper are available on the website (https://zenodo.org, doi:10.5281/zenodo.6244759).

*Author contributions*. YL performed the data analysis and prepared the manuscript. MPC and WF performed the model simulations. SSD, MPC, WF, AC, YX and DG gave support for discussion, simulation and interpretation, and helped to write the paper. All authors edited and contributed to subsequent drafts of the manuscript.

*Competing interests*. The authors declare that they have no conflicts of interest.

*Acknowledgements*. We are grateful to the Copernicus Climate Change Service (C3S) for providing the global ozone dataset. The modelling work is supported by National Centre for Atmospheric Science (NCAS). We thank all providers of the climate data used in this study.

*Financial Support*. We acknowledge the support of the National Natural Science Foundation of China (grant no. 41675039,

91837311), the Natural Science Foundation for universities in Jiangsu province (grant no. 20KJD170001, 21KJB510007) and the Scientific Research Project of Nanjing Xiaozhuang University (grant no. 2019NXY42). This research is partially supported by the Natural Environment Research Council (grant no. NE/R001782/1), the National Centre for Earth Observation (grant no. NE/R016518/1) and NCAS (NE/R015244/1).

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
