# Peer review of "Effects of Reanalysis Forcing Fields on Ozone Trends and Age-of-Air from a Chemical Transport Model"

_Atmospheric Chemistry and Physics, 2022_

## Author Comment (AC1)

**Reply to Reviewer #1:**

We thank the reviewer for his/her useful comments and suggestions which have helped to improve the manuscript. The reviewer's comments are given below in black text, followed by our responses in blue text.

We believe that major revisions suggested by the reviewer are as follows:
1) Rewrite Section 2.3: use independent linear trends (ILT) in the regression model.
2) Adjust the structure: add sub-subsections to Section 3.1 and Section 3.2.
3) Update the results: re-plot the linear trends using ILT in Sections 3.1.3 and 3.2.3.
4) Add figures to compare simulations with observations: see Figure S1, Figure 6 and Figure S3.
5) Modify/Strengthen the AoA part: see the title, Abstract, Introduction, Section 2.2 and Section 4.

Our responses on these points are given below.

Li et al. compare in their manuscript ozone, simulated with the CTM TOMCAT, based on the reanalyses ERA-Interim and ERA5 with different observational collections for total column ozone and ozone profiles in the time frame 1979-2018. They find that ERA5 better reproduces measured TCO anomalies, while trends and profiles are better reproduced by ERA-Interim.

The paper discusses an important topic, which is certainly within the scope of ACP. However, major revisions are necessary before publication in ACP. In particular the structure of the paper should be revised to have it more concise, with some of the current figures moved into a supplement. In addition, some background information of the usage of the TOMCAT model and the used methods are missing. In the second part of the paper, it is a major weakness that the observational SWOOSH data is barely used. The AoA part seems to be not connected to the preceding part of the paper.

General points:

 - In the methods, the PWLT regression model is introduced, but later in the analyses it is not mentioned in which way the data shown has been treated by this model. Probably, the model has been used to calculate the anomalies, but this should be mentioned.

Reply: We agree with the reviewer and following Reviewer #2's suggestion, we have rewritten Section 2.3 using independent linear trends (ILT) to replace PWLT (ILT is similar to the PWLT but with additional step-function-like term that allows trend terms to be independent of each other). We also clarify that the regression model uses absolute values, and have revised that description of the regression model to avoid confusion.

 - Section 3.1 is lengthy for my taste. Information from some of the figures is repetitive, so maybe some of the figures could be moved to a supplement? Further, the structure of this section is not very clear to me. Maybe it could help to have sub-subsections? Or maybe it is sufficient to shorten this part?

Reply: We have moved the original Figure 4 to the supplement (Figure S2). We also added sub-subsections to make the structure clear:
3.1.1 Total column ozone anomalies
3.1.2 Seasonal variability in total column ozone
3.1.3 Total column ozone trends and explanatory variables

- In Section 3.2, the observational SWOOSH data is barely used in this article. Instead the profile information of both model runs is compared with each other. It would be more helpful to assess which reanalysis data set performs better by comparisons to the observations.

Reply: We have added a figure in the manuscript (Figure 6) to compare differences in ozone anomalies between two simulations and the observational SWOOSH data. We also compared the stratospheric column ozone (SCO) integrated from 316 hPa to 1 hPa based on SWOOSH and model simulations (Figure S1) and the profiles of correlation coefficients between simulations and SWOOSH data (Figure S3). Overall, our analysis suggests that although ERA5-based simulations show better agreement with observation-based datasets, they do show, however, larger biases over certain regions (e.g. NH in winter). Hence, it is very difficult to argue that ERA5 is better than ERA-Interim.

- Similar to the comment on Section 3.1, Section 3.2 would benefit from more external structure.

Reply: We added sub-subsections in this section to make the structure clear:
3.2.1 Comparison of vertical ozone profiles
3.2.2 Interannual variability of ozone and temperature anomalies
3.2.3 Comparison of ozone profile trends

- It would be very helpful to know how the two different observational data sets compare to each other, since the outcome of the comparisons to the simulation results are quite different. I think it would be a good idea to calculate TCO from the SWOOSH profiles and include these data in the comparisons of Section 3.1 (or in the supplement) to show to which degree these observations are consistent.

Reply: As we cannot obtain TCO from SWOOSH profiles, we calculate the stratospheric column ozone (SCO) integrated from 316 hPa to 1 hPa based on SWOOSH and model simulations over the period 1984-2018 as shown in the supplementary Figure S1. We find that both simulations can reproduce the seasonal characteristics of SCO from SWOOSH although they are overestimated for most mid-high latitude regions. The comparison of SCO between the two simulations shows consistent results with those for TCO in Figure 2f.

[Figure]

Figure S1: Zonal and monthly mean stratospheric column ozone (SCO, in DU) climatology over the period 1984-2018 based on (a) SWOOSH, (b) A_ERAI and (c) B_ERA5. The absolute differences between each simulation and SWOOSH, as well as between the two simulations, are shown in (d) A_ERAI - SWOOSH, (e) B_ERA5 - SWOOSH and (f) B_ERA5 - A_ERAI, respectively.

- Section 3.3 seems to be a related, but different topic. It is also not reflected by the title of this paper. I would suggest to remove this AoA part from this manuscript, and rather focus on the ozone trends. Otherwise a lot of more work would be necessary to properly introduce the measurement data sets used to compare the model with. However, this would result in an even longer paper, which may be less focused.

Reply: We thank the reviewer for this suggestion. However, we decided to keep this AoA discussion because we feel that this is a straightforward and useful diagnostic that helps to understand key differences in the stratospheric transport. We have tried to strengthen this part of the manuscript and at the same time make sure the paper is as concise as possible. Detailed modifications involving the title, Abstract, Introduction, Section 2.2  and Section 4 are marked in red in the revised manuscript.

Specific points:

Introduction:

- Line 45: "Besides the decrease in ODSs, cooling induced by increased greenhouse gases (GHGs) slows the rate of ozone loss [...]": This view is a little too simplistic, as it is known that cooler stratospheric temperatures may lead to larger PSC areas and thus more catalytic ozone loss (see e.g., Rex et al., 2004, 10.1029/2003GL018844)

Reply: Yes, we agree, but our statement is largely true for global ozone. Also, increasing GHGs are expected to increase strength of the stratospheric circulation thereby weakening the polar vortex (or

isolated region that permit PSC formation). Cooler wintertime stratospheric temperatures with the polar vortices do lead to larger PSC areas and thus more catalytic ozone loss (e.g. Solomon, 1999; Rex et al., 2004). In the upper stratosphere where photochemical processes control the ozone level, there is broad agreement in that ozone increases as ODSs decrease and temperature decreases due to greenhouse gas increase (e.g. Douglass et al., 2012).

We modified the sentence with supplementary information (Lines 46-51 in the revised manuscript): "Besides the decrease in ODSs, the increased greenhouse gas (GHG) abundances warm the troposphere, causing strengthening of the stratospheric circulation and increases in tropical upwelling, which reduces ozone in the tropical lower stratosphere (e.g. Bekki et al., 2013; Marsh et al., 2016). Also, increasing GHGs cause stratospheric cooling that slows gas-phase ozone loss cycles and is expected to speed-up recovery in upper stratospheric ozone globally (e.g. Haigh and Pyle, 1982; Eyring et al., 2010; Douglass et al., 2012)."

*Related references:*

*Douglass, A. R., Stolarski, R. S., Strahan, S. E., Oman, L. D.: Understanding differences in upper stratospheric ozone response to changes in chlorine and temperature as computed using CCMVal-2 models, J. Geophys. Res., 117(D16), 16306, 2012.*

*Eyring, V., Cionni, I., Lamarque, J. F., Akiyoshi, H., Bodeker, G. E., Charlton-Perez, A. J., Frith, S. M., Gettelman, A., Kinnison, D. E., Nakamura, T., and Oman, L. D.: Sensitivity of 21st century stratospheric ozone to greenhouse gas scenarios. Geophys. Res. Lett., 37(16), doi:10.1029/2010GL044443, 2010.*

*Haigh, J. D. and Pyle, J. A.: Ozone perturbation experiments in a two-dimensional circulation model, Q. J. R. Meteorol. Soc., 108, 551-574, doi:10.1002/qj.49710845705, 1982.*

*Rex, M., Salawitch, R. J., von der Gathen, P., Harris, N. R. P., Chipperfield, M. P., and Naujokat, B.: Arctic ozone loss and climate change, Geophys. Res. Lett., 31, L04116, doi:10.1029/2003GL018844, 2004.*

*Solomon, S.: Stratospheric ozone depletion: A review of concepts and history. Reviews of Geophysics, 37(3):275–316, 1999.*

Section 2.1:

 - It is noted that the TOMCAT CTM uses ODSs, GHGs, aerosols from volcanic eruptions, but it is not mentioned how these variables are initialized, and how emissions are handled.

Reply: We added some information about the emissions and variables used in TOMCAT (Lines 113-117): "For major ODSs and GHGs, the model uses updated global mean surface mixing ratio scenarios (Carpenter et al., 2018) which are treated as well mixed throughout the troposphere. The implementation of sulphate aerosol surface area density (SAD) and solar flux variations are described in Dhomse et al. (2015; 2016). Aerosol (or SAD) variations from volcanic eruptions are from Luo (2016), whereas solar flux variations are taken from the NRL2 empirical model (Coddington et al., 2016)."

*Related references:*
*Carpenter, L. J., Daniel, J. S., Fleming, E. L., Hanaoka, T., Ju, H., Ravishankara, A. R., Ross, M. N., Tilmes, S., Wallington, T. J., and Wuebbles, D. J.: Scenarios and information for policy makers, in: Scientific Assessment of Ozone Depletion: 2018, World Meteorological Organization, Global Ozone*

*Research and Monitoring Project – Report No. 58, chap. 6, World Meteorological Organization/ UNEP, Geneva, Switzerland, 2018.*

*Coddington, O., Lean, J. L., Pilewskie, P., Snow, M., and Lindholm, D.: A solar irradiance climate data record, B. Am. Meteorol. Soc., 97, 1265–1282, doi: 10.1175/BAMS-D-14-00265.1, 2016.*

*Dhomse, S. S., Chipperfield, M. P., Feng, W., Hossaini, R., Mann, G. W., and Santee, M. L.: Revisiting the hemispheric asymmetry in midlatitude ozone changes following the Mount Pinatubo eruption: A 3-D model study, Geophys. Res. Lett., 42, 3038-3047, doi:10.1002/2015gl063052, 2015.*

*Dhomse, S. S., Chipperfield, M., Damadeo, R., Zawodny, J., Ball, W., Feng, W., Hossaini, R., Mann, G., and Haigh, J.: On the ambiguous nature of the 11 year solar cycle signal in upper stratospheric ozone, Geophys. Res. Lett., 43, 7241–7249, doi:10.1002/2016GL069958, 2016.*

*Luo, B.: Stratospheric aerosol data for use in CMIP6 models, available at: ftp://iacftp.ethz.ch/pub_read/luo/CMIP6/Readme_Data_Description.pdf (last access: 1 June 2021), 2016.*

 - For the TOMCAT run with ERA5, it is not stated how the considerably higher resolved ERA5 data is degraded to the lower resolved TOMCAT horizontal and vertical resolution. It should be further discussed, how this downsampling could influence the comparison with ERAI, which has a more similar horizontal and vertical resolution as TOMCAT.

- Further, it is not clear to me, if for the ERA5 run, the hourly output of ERA5 has been used. Again, the influence of different temporal sampling from the driving fields should be discussed.

Reply: The model grid is variable and meteorological fields are obtained by converting the forcing ECMWF meteorological analyses to grid-point fields using a spectral transform. For more details please see Chipperfield (2006). TOMCAT generally reads in temperature, winds and surface pressure as spectral coefficients. If the resolution of forcing fields is higher than the model grid, then information from the higher wavenumbers is not used and the spectral coefficients are truncated (Feng et al., 2011).

In our manuscript, both CTM runs (A_ERAI and B_ERA5) are forced using ECMWF analyses at T42 resolution ($2.8^{\circ} \times 2.8^{\circ}$). The hourly output of ERA5 is not used. Instead, the 6-hourly grid point meteorological fields are interpolated linearly in time for both runs. Although ERAI and ERA5 data are degraded to the lower resolved TOMCAT resolutions, the primary focus of this study is to analyze the long-term large-scale variations in ozone concentrations. Some of the earlier studies such as Feng et al. (2011), Grooß et al., 2018 did investigate the effect of resolution ranging from $5.6° \times 5.6°$ to $1.1° \times 1.1°$ and found that the higher model resolution does not show significant variations in model simulated tracers. Additionally, we are also constrained by the computational resources in terms of storing and using hourly data.

We also included some additional information above in the revised manuscript (Lines 141-147): "The model grid is variable and determined by converting the forcing ECMWF meteorological analyses to grid-point fields using a spectral transformation (Chipperfield, 2006). Both TOMCAT simulations are performed at $2.8° \times 2.8°$ (T42 Gaussian grid) horizontal resolution and have 32 hybrid sigma-pressure levels ranging from the surface to about 60 km. The 6-hourly grid point meteorological fields are interpolated linearly in time for both runs. Although ERA5 provides hourly output including information about uncertainties, here we used 6-hourly fields as ERA-Interim provides output at the same frequency and to reduce storage requirements."

*Related references:*
*Chipperfield, M. P.: New version of the TOMCAT/SLIMCAT off-line chemical transport model: Intercomparison of stratospheric tracer experiments, Q. J. Roy. Meteorol. Soc., 132, 1179-1203, doi:10.1256/qj.05.51, 2006.*
*Feng, W., Chipperfield, M. P., Dhomse, S., Monge-Sanz, B. M., Yang, X., Zhang, K., and Ramonet, M.: Evaluation of cloud convection and tracer transport in a three-dimensional chemical transport model, Atmos. Chem. Phys., 11, 5783–5803, doi: 10.5194/acp-11-5783-2011, 2011.*
*Grooß, J.-U., Müller, R., Spang, R., Tritscher, I., Wegner, T., Chipperfield, M. P., Feng, W., Kinnison, D. E., and Madronich, S.: On the discrepancy of HCl processing in the core of the wintertime polar vortices, Atmos. Chem. Phys., 18, 8647–8666, doi: 10.5194/acp-18-8647-2018, 2018.*

 - It would be interesting to know what kind of observations have been assimilated to ERAI and ERA5. I guess they also both used some kind of ozone information to get radiation and dynamics.

Reply: Detailed information on observations that have been assimilated to ERAI and ERA5 can be found in Hersbach et al. (2020). Modified texts in the revised manuscript (Lines 125-136) are pasted below: "Although almost all of the radiance datasets assimilated in ERA-Interim are included in ERA5, an updated radiative transfer model is used in ERA5 (RTTOVS-v11 against RTOVS v7) that includes several developments and various reprocessed radiance datasets (see Figure 5 in Hersbach et al., 2020). Major differences between two reanalysis datasets also include significant divergence in terms of volume of radiance measurements assimilated post-2007 (more and newer observations in ERA5), which are not assimilated in ERA-Interim, together with the gradual decline in the numbers assimilated in ERA-Interim, as instruments and channels gradually failed. In both ERA-Interim and ERA5 reanalysis systems, the prognostic ozone model is based on the parameterization scheme of Cariolle and Teyssèdre (2007). Simply, both systems have the ozone evolution that is expressed as a linear expansion with respect to the photochemical equilibrium for the local values of the ozone mass mixing ratio, the overhead ozone column, the temperature and some ozone depletion during polar winter. The ozone tracer is advected with the model flow. It is important to note that the ERA5 prognostic ozone is not coupled with the radiation scheme which uses diagnostic time varying ozone fields recommended for CMIP5 simulations."

*Related references:*
*Cariolle, D. and Teyssèdre, H.: A revised linear ozone photochemistry parameterization for use in transport and general circulation models: multi-annual simulations, Atmos. Chem. Phys., 7, 2183–2196, doi: 10.5194/acp-7-2183-2007, 2007.*
*Hersbach, H., Bell, B., Berrisford, P., Hirahara, S., Horányi, A., Muñoz-Sabater, J., Nicolas, J., Peubey, C., Radu, R., Schepers, D., Simmons, A., Soci, C., Abdalla, S., Abellan, X., Balsamo, G., Bechtold, P., Biavati, G., Bidlot, J., Bonavita, M., De Chiara, G., Dahlgren, P., Dee, D., Diamantakis, M., Dragani, R., Flemming, J., Forbes, R., Fuentes, M., Geer, A., Haimberger, L., Healy, S., Hogan, R. J., Hólm, E., Janisková, M., Keeley, S., Laloyaux, P., Lopez, P., Lupu, C., Radnoti, G., de Rosnay, P., Rozum, I., Vamborg, F., Villaume, S., and Thépaut, J.-N.: The ERA5 global reanalysis, Q. J. Roy. Meteorol. Soc., 146, 1999-2049, doi:10.1002/qj.3803, 2020.*

Section 2.2:

- The URL to the C3S data directs to a page which states, that it should not be used any more. Instead this data set should be used: https://cds.climate.copernicus.eu/cdsapp#!/dataset/satellite-ozone-v1?tab=overview. It looks like, the authors used the data set with known bugs for this manuscript. The authors should discuss these known bugs and how these are affecting the analysis.

Reply: We thank the reviewer for the reminder. We have checked the data and found few differences between the two versions. Fortunately, these known bugs do not affect the results and analysis in our manuscript. We have updated the data website in the revised manuscript (Line 150).

- For the SWOOSH data some information of vertical resolution would be helpful, in particular in comparison to the vertical resolution of TOMCAT.

Reply: We added some information on the vertical and horizontal resolution for SWOOSH data in the revised manuscript (Line 170): "The merged SWOOSH record spans from 1984 to present, and consists of monthly mean zonal-mean ozone values at grids of 2.5° and 12 levels per decade in pressure from 316 to 1 hPa (31 pressure levels)."

Section 2.3:

- Equation (1) is written down in a confusing way. After reading all the text, it becomes clear, why there are two linear terms, but it would be better to define the function piecewise instead (see https://en.wikipedia.org/wiki/Piecewise).

Reply: Sorry for the confusion. We have rewritten this part. As the piecewise linear trend (PWLT), also called hockey stick, is a common method used for analyzing the ozone trends, we cited some references here. In order to avoid further confusion and as suggested by the second reviewer, we modified the regression method using independent linear trends (ILT). In the revised manuscript and updated supplementary material, we have expanded the description of the regression methods in Section 2.3.

- I guess the proxy functions have been used earlier. It would be helpful to have a reference to a publication, where the reader could look up the $X_i(t)$ functions used for this work.

Reply: Yes. We cited some references here. As these proxy functions have been extended with new data, we also provided the sources of the updated proxy series (see Section 2.3).

- Line 154: Typo "Steinbrecht et al. 2017"

Reply: Corrected.

- Line 166: Missing subscript in Xi

Reply: We have added the subscript.

Section 3:

- I do not see the point of the text here. It reads like an introduction to Section 3.1, so it could be moved there directly.

Reply: To avoid confusion, we made some adjustment. Some texts related to the regression method are moved to Section 2.3, some related to the regression results of total column ozone are moved to Section 3.1.3 and some related to the trend results of vertical ozone profiles are move to Section 3.2.3.

Section 3.1:

 - Figure 1 (caption): Please also mention that these are monthly means displayed here.

Reply: Added the "Monthly mean" in the figure caption.

 - Line 182: "Anomalies are calculated by subtracting the long-term monthly average from each monthly mean value.": Does that mean that $C_0$ from equation (1) is subtracted? Otherwise, how is the PWLT regression model used here?

Reply: Sorry for the confusion. The PWLT regression model is not used here. As described, anomalies are calculated by subtracting the long-term monthly average from each monthly mean value. They are not obtained from the equation (1).

 - Line 193: "[...] obtained from merged C3S and TOMCAT simulations, A_ERAI and B_ERA5 [...]": For me, this reads like C3S and TOMCAT simulation data are merged in some way, but the figure looks more like these individual data sets are displayed together in the left panels of Fig.1.

Reply: We deleted the "merged" here.

 - Line 208: "[...] such as representation of dynamical processes in the ERA-Interim reanalyses.": Please give some examples of such dynamical processes.

Reply: We modified the sentence with some examples and references related to dynamical processes (Lines 232-236): "The relatively better agreement between B_ERA5 and C3S in the NH mid-high-latitude regions could be due to improvements in the representation of dynamical processes in ERA5 reanalysis data, such as convection in the upper troposphere and lower stratosphere (Li et al., 2020a) and residual mean mass circulation of the BDC in the stratosphere (Diallo et al., 2021; Ploeger et al., 2021)."

*Related references:*

*Diallo, M., Ern, M., and Ploeger, F.: The advective Brewer– Dobson circulation in the ERA5 reanalysis: climatology, vari- ability, and trends, Atmos. Chem. Phys., 21, 7515-7544, doi: 10.5194/acp-21-7515-2021, 2021.*

*Li, D., Vogel, B., Müller, R., Bian, J., Günther, G., Ploeger, F., Li, Q., Zhang, J., Bai, Z., Vömel, H., and Riese, M.: Dehy- dration and low ozone in the tropopause layer over the Asian monsoon caused by tropical cyclones: Lagrangian transport calculations using ERA-Interim and ERA5 reanalysis data, Atmos. Chem. Phys., 20, 4133-4152, doi: 10.5194/acp-20-4133-2020, 2020a.*

*Ploeger, F., Diallo, M., Charlesworth, E., Konopka, P., Legras, B., Laube, J. C., Grooß, J. U., Günther, G., Engel, A., and Riese, M.: The stratospheric Brewer–Dobson circulation inferred from age of air in the ERA5 reanalysis, Atmos. Chem. Phys., 21, 8393-8412, doi: 10.5194/acp-21-8393-2021, 2021.*

- Figure 2: Color bars are missing.

Reply: Corrected.

- Figure 2 and corresponding text: Again, it is not clear to me, how the PWLT regression model has been used here, as it was stated in the beginning of Section 3.

Reply: There is no PWLT regression model for Figure 2. It might be possible that reviewer's comments were targeted towards Figure 4. We have modified the structure of the paper and the discussion about regression methods is expanded in Section 2.3.

- Line 221: "[...] B_ERA5 shows larger positive biases (more than 15 DU) during NH winter-spring seasons [...]": Why is that so? It seems like Arctic ozone depletion is underestimated here. Are the Arctic vortices not represented well in ERA5 (and ERAI in a less pronounced way)?

Reply: We believe the Arctic vortices are represented well in ERA5. Feng et al. (2021) analyzed the Arctic ozone depletion in 2019/20 using TOMCAT forced with ERA5 reanalysis and showed that TOMCAT captures well the observed persistent low temperatures and extensive chlorine activation. Simulation B_ERA5 shows larger positive biases in the NH winter-spring seasons. This problem arises from the fact that ozone is primarily produced in the tropics and is transported to the mid-high latitudes via stratospheric circulation. The differences with simulation A_ERAI indicate that transport pathways between two reanalysis systems are different from the expected behaviour such as increased transport decreasing tropical ozone and increasing mid-high latitude ozone (e.g. Weber et al., 2003; Dhomse et al., 2006; Chrysanthou et al., 2019). We added some discussion in the revised manuscript (Lines 262-265).

*Related references:*
*Chrysanthou, A., Maycock, A. C., Chipperfield, M. P., Dhomse, S., Garny, H., Kinnison, D., Akiyoshi, H., Deushi, M., Garcia, R. R., Jöckel, P., Kirner, O., Pitari, G., Plummer, D. A., Revell, L., Rozanov, E., Stenke, A., Tanaka, T. Y., Visioni, D., and Yamashita, Y.: The effect of atmospheric nudging on the stratospheric residual circulation in chemistry–climate models, Atmos. Chem. Phys., 19, 11559–11586, doi: 10.5194/acp-19-11559-2019, 2019.*

*Dhomse, S. S., Weber, M., Wohltmann, I., Rex, M., and Burrows, J. P.: On the possible causes of recent increases in northern hemispheric total ozone from a statistical analysis of satellite data from 1979 to 2003, Atmos. Chem. Phys., 6, 1165-1180, doi:10.5194/acp-6-1165-2006, 2006.*

*Feng, W., Dhomse, S. S., Arosio, C., Weber, M., Burrows, J. P., Santee, M. L., and Chipperfield, M. P.: Arctic ozone depletion in 2019/20: Roles of chemistry, dynamics and the Montreal Protocol, Geophys. Res. Lett., 48, e2020GL091911, doi: 10.1029/2020GL091911, 2021.*

*Weber, M., Dhomse, S., Wittrock, F., Richter, A., Sinnhuber, B.-M., and Burrows, J. P.: Dynamical control of NH and SH winter/spring total ozone from GOME observations in 1995–2002, Geophys. Res. Lett., 30, doi:10.1029/2002GL016799, 2003.*

- Line 239: "Both CTM simulations capture the observed seasonal characteristics of TCO variations averaged across all latitude bands considered here": How can one make such a statement based on DJF and JJA timeseries over several years?

Reply: We modified the sentence with "Figure 3 compares the TCO evolution during DJF and JJA seasons averaged over 60°N-35°N, 20°N-20°S and 35°S-60°S from C3S, A_ERAI and B_ERA5, and their differences are shown in the supplementary Figure S2. TCO variations from both CTM simulations show reasonable agreement with C3S data, in line with the results in Figure 2." (Lines 280-282).

 - Figure 3: The caption should warn the reader that y-axes change for the different panel rows.

Reply: We added the information "Note that y-axes change for different panels." in the caption (Line 279).

 - Figure 3: Are the authors really sure about their correlation coefficients? For panel 3e, the red line seems to be further away from the black line almost for the whole time series than the blue line, but its correlation coefficients is said to be better. Also, for the other panels, the rather good correlation coefficients look strange to me. But maybe the offset is confusing my eyes.

Reply: We have checked our correlation coefficients and there is no issue. We use the Pearson correlation coefficient which is most widely used for the measurement of linear relationship between two datasets. As to the larger biases but better correlation coefficients in simulation B_ERA5, we have some discussion in the text (Lines 299-301): "Despite larger biases in B_ERA5 than in A_ERAI (e.g. Figures 3a and 3e), the correlation coefficients between B_ERA5 and C3S are higher than A_ERAI, which suggests that A_ERAI might have some unrealistic annual variability."

 - Discussion of Figure 3: Both models are overestimating DJF O3 in the north, and in particular ERA5 is underestimating DJF O3 in the south. Please discuss.

Reply: Both models overestimate DJF mean TCO in the north, which might be due to the greater wintertime transport estimated in model simulations, as the poleward transport of ozone is most effective in the winter hemisphere (e.g. Chipperfield and Jones, 1999). B_ERA5 overestimates the observed DJF mean TCO in the NH mid-latitudes while it underestimates it in the SH mid-latitudes, which might be due to the differences in simulated wintertime ozone build-up (transport dominates) and summertime ozone losses (photochemical loss dominates) compared to observations. We added some discussion above in the revised manuscript (Lines 293-298).

*Related references:*
*Chipperfield, M. P., and Jones, R. L:. Relative influence of atmospheric chemistry and transport on Arctic ozone trends, Nature, 400, 551-554, doi:10.1038/22999, 1999.*

 - Figure 4 caption: "The absolute differences with the standard deviations averaged over the whole period between simulation A_ERAI (B_ERA5) and C3S are presented in blue (red) text": This is confusing. Maybe the authors mean: "The absolute differences averaged over the whole period between simulation A_ERAI (B_ERA5) and C3S are presented together with the standard deviations in blue (red) text."?

Reply: We corrected this sentence as suggested (Figure S2).

 - Discussion of Figure 4: This is very repetitive compared to the discussion of Figure 3. I suggest to move one of the figures to the supplement and bring both discussions together.

Reply: We moved the original Figure 4 to the supplement (Figure S2) and brought both discussions together. More detailed modifications are marked in red in the revised manuscript (Lines 287-293).

- Line 277: "The regression model used here is identical to that used in Li et al. (2020), except for the different explanatory variables considered for different latitude bands": Is this regression model also identical to the one described in Section 2.3? In that case, this sentence should be moved there, and in Section 3.1, it should be referred to Section 2.3. Otherwise, this different regression model should be introduced in more detail in a new section 2.4.

Reply: We moved the sentence "The regression model used for the total column ozone in December-January-February (DJF) and June-July-August (JJA) seasons are identical to that used in Li et al. (2020b)." to Section 2.3 (Lines 204-205). We referred this part to Section 2.3. We also added sub-subsections (Section 3.1.3) to make the structure clear.

- Table 1: The information of this table could be integrated into Figure 5 (similar to the regression coefficients in Figures 1 or 3).

Reply: We integrated the information of this table (determination coefficients of the regression model) in the new Figure 4 in which independent linear trends (ILT) are used to replace PWLT and proxies of aerosol, AO and AAO terms are removed to avoid the significant correlation coefficients with other proxies.

- Line 297: "[...] with more significant decreases at NH and SH mid-latitude bands for the simulations than C3S.": Does that mean that there is a general problem in TOMCAT or in both reanalyses?

Reply: There is no clear answer. TOMCAT chemical scheme and representation of dynamical processes might have some issues but we are using the identical chemical scheme in both the simulations. So, some identical differences (absolute differences in the upper and lower stratosphere) can be due to biases in ozone production and loss processes. However, time varying ozone differences can be directly linked to the dynamical forcings. On the other hand, observational data sets are also subject to errors associated with the measurement techniques, instrument degradation and retrieval algorithms. Hence, almost all chemical models may be expected to show a bias against observational data records, either because of model deficiencies or errors in the observations. Here, the more significant ozone decreases before 1998 for mid-latitude bands in simulations than in C3S suggests that the incomplete presentation of complex atmospheric processes and photochemical parameters applied in the reanalysis and model, and uncertainties in observational data sets are possibly responsible for this difference.

Modified texts in the revised manuscript (Lines 313-316) are pasted below: "The overestimation of these negative trends in model simulations could be due to (1) the unrealistic trends in stratospheric transport, especially during 1980s and 1990s, (2) the incomplete presentation of complex atmospheric processes and their feedbacks, (3) the incorrect parameterization for photochemical reactions in a CTM or (4) the uncertainties in observational data sets (e.g. Dhomse et al., 2021)."

*Related references:*

*Dhomse, S. S., Arosio, C., Feng, W., Rozanov, A., Weber, M., and Chipperfield, M. P.: ML-TOMCAT: machine-learning-based satellite-corrected global stratospheric ozone profile data set from a*

*chemical transport model, Earth Syst. Sci. Data, 13, 5711–5729, doi:10.5194/essd-13-5711-2021, 2021.*

- Line 299: "The recovery since 1998 (Trend2) from C3S is quite different to that from the simulations in terms of its magnitude and significance.": This is not really true for the tropics, where Trend2 seems to agree between C3S and ERA5.

Reply: We have used the ILT method to replace PWLT, but we find the updated results are generally consistent with those using PWLT. To be more clear, we modified this sentence in the revised manuscript (Lines 325-326): "The recovery since 1998 (Trend2) from C3S and both simulations differs from each other in terms of its magnitude and significance". More detailed modifications are marked in red in the manuscript.

Section 3.2

- Figure 6: I got confused by the titles "1984-2018" for all panels but the first column. Since in all of the panels the same time range is shown, this information could be replaced by a more meaningful title (e.g. "Rel. Diff to SWOOSH" in column 2).

Reply: We replotted the figure (the new Figure 5) and used "Rel. Diff to SWOOSH" as the title.

- ERA5 seems to be a bit worse than ERAI for all latitudinal bands, except for the southern UTLS region. But in general, both TOMCAT runs agree in underestimating the upper stratosphere, and overestimating the lower stratosphere. Can the authors exclude that this is a general problem of TOMCAT? How much is the vertical resolution of the model, since it only has 32 vertical levels?

Reply: In general, both TOMCAT runs agree in underestimating the upper stratosphere, and overestimating the lower stratosphere. This structure might be associated with deficiencies in the photochemical reactions and dynamical processes in the model (Dhomse et al., 2021). The negative ozone biases in the upper stratosphere are most probably due to uncertainties in the solar irradiances and/or photolysis cross sections (e.g. Brasseur and Solomon, 2006) and uncertainties in temperature-dependent reaction rates (e.g. Ghosh et al., 1997; Stolarski et al., 2010; Dhomse et al., 2013, 2016). The positive biases in the lower stratosphere could be due to a combination of both dynamics and chemistry. The incomplete representation of key physical processes and of various circulation pathways in the lower stratosphere (e.g. downward transport controlled by the QBO, and stratospheric transport determined by the strength of the Brewer–Dobson circulation) in the ERA5 reanalysis scheme could impact the meteorology used in the CTM (e.g. Mitchell et al., 2020; Dhomse et al., 2021).

Besides, model resolution could also influence the performance of model simulations (Wang et al., 2019). TOMCAT ozone profiles (with 32 hybrid sigma-pressure levels) are linearly interpolated in log-pressure on to 43 equidistance pressure levels (1000-0.1 hPa), with the same coverage from 316 to 1 hPa as SWOOSH ozone profiles.

We also added some discussion in the revised manuscript (Lines 347-348): "These biases might be associated with deficiencies in the representation of the photochemical reactions and dynamical processes in the model (e.g. Mitchell et al., 2020; Dhomse et al., 2013; 2016; 2021).".

*Related references:*

*Brasseur, G. P. and Solomon, S.: Aeronomy of the middle atmosphere: Chemistry and physics of the stratosphere and mesosphere, vol. 32, Springer Science & Business Media, ISBN 978- 1-4020-3284-4, doi: 10.1007/1-4020-3824-0, 2006.*

*Dhomse, S. S., Chipperfield, M. P., Feng, W., Ball, W. T., Unruh, Y. C., Haigh, J. D., Krivova, N. A., Solanki, S. K., and Smith, A. K.: Stratospheric O3 changes during 2001–2010: the small role of solar flux variations in a chemical transport model, Atmos. Chem. Phys., 13, 10113–10123, doi: 10.5194/acp-13- 10113-2013, 2013.*

*Dhomse, S. S., Chipperfield, M. P., Damadeo, R., Zawodny, J., Ball, W., Feng, W., Hossaini, R., Mann, G., and Haigh, J.: On the ambiguous nature of the 11 year solar cycle signal in upper stratospheric ozone, Geophys. Res. Lett., 43, 7241–7249, doi:10.1002/2016GL069958, 2016.*

*Dhomse, S. S., Arosio, C., Feng, W., Rozanov, A., Weber, M., and Chipperfield, M. P.: ML-TOMCAT: machine-learning-based satellite-corrected global stratospheric ozone profile data set from a chemical transport model, Earth Syst. Sci. Data, 13, 5711–5729, doi:10.5194/essd-13-5711-2021, 2021.*

*Ghosh, S., Pyle, J. A., and Good, P.: Temperature dependence of the ClO concentration near the stratopause, J. Geophys. Res.- Atmos., 102, 19207–19216, doi: org/10.1029/97JD01099, 1997.*

*Mitchell, D. M., Lo, Y. E., Seviour, W. J., Haimberger, L., and Polvani, L. M.: The vertical profile of recent tropical temperature trends: Persistent model biases in the context of internal variability, Environ. Res. Lett., 15, 1040b4, doi: 10.1088/1748-9326/ab9af7, 2020.*

*Stolarski, R. S., Douglass, A. R., Newman, P. A., Pawson, S., and Schoeberl, M. R.: Relative Contribution of Greenhouse Gases and Ozone-Depleting Substances to Temperature Trends in the Stratosphere: A Chemistry–Climate Model Study, J. Climate, 23, 28–42, doi: 10.1175/2009JCLI2955.1, 2010.*

*Wang, W., Shangguan, M., Tian, W., Schmidt, T., & Ding, A.: Large uncertainties in estimation of tropical tropopause temperature variabilities due to model vertical resolution. Geophysical Research Letters, 46, 10043-10052. doi: 10.1029/2019GL084112, 2019.*

- Figure 7: Why are the models not compared to the measurements over time? I find it hard to judge which model performed better concerning the profile just by comparing profiles averaged over 34 years! In Fig. 6 it can be seen that both simulations having a hard time to reproduce the measurements. It would be helpful to know if this is an overall problem or limited to a certain period! Figures S1 and S2 are more important than Figure 7!

Reply: We added a new figure (Figure 6) in the revised manuscript, which is combined from the original Figures S1 and S2, to compare the modelled stratospheric ozone anomalies with the SWOOSH ozone anomalies over the time period 1984-2019 (August).

The text related to this new Figure 6 is correspondingly (Lines 369-381): "After analyzing biases in mean ozone profiles, we diagnose the time-dependent differences in ozone anomalies between two simulations and SWOOSH data set as shown in Figure 6. The relative differences between simulated and observed ozone anomalies are calculated with respect to SWOOSH ozone values. The comparison shows that simulation A_ERAI significantly overestimates the observed NH mid-latitude ozone anomalies for the early years (1984-1996) over the whole stratosphere especially in the lowermost stratosphere. Afterwards ozone anomalies in A_ERAI are underestimated, while ozone anomalies in B_ERA5 are more comparable with the observations except for the significant

overestimation in the lower stratosphere during the later period 2006-2019 (August). The situation in the SH mid-latitude region is similar to that in the NH except that the biases are relatively smaller. These results are consistent with the comparison of TCO anomalies shown in Figures 1g and 1i, also indicating that differences in the lower stratosphere are mainly responsible for their differences in TCO. In the tropics, both simulations underestimate the observed ozone anomalies in the lower stratosphere before 2000 but overestimate them afterwards. However, in the upper stratosphere (above 3 hPa) cases are opposite for the two simulations, which might be associated with the uncertainties in temperature-dependent reaction rates in the models (e.g. Stolarski et al., 2010; Dhomse et al., 2013, 2016)."

[Figure]

Figure 6: Pressure-time cross section of the relative differences (%) in ozone anomalies between model simulations A_ERAI, B_ERA5 and SWOOSH over 1984-2019 (August) for different latitude regions (a, b) 60°N-35°N, (c, d) 20°N-20°S and (e, f) 35°S-60°S.

And "We also compare the profiles of correlation coefficients between the simulated and SWOOSH ozone anomalies over the latitude bands 60°N-35°N, 20°N-20°S and 35°S-60°S, as shown in Figure S3. Again, though simulation B_ERA5 generally shows better correlation with C3S TCO anomalies (Figure 1), it does not show better correlation for the stratospheric ozone profile anomalies overall (e.g. in the upper stratosphere for the tropics and NH mid-latitudes and in the lower stratosphere for the SH mid-latitudes). " (Lines 381-385).

[Figure]

Figure S3: Correlation coefficients between the simulated and SWOOSH ozone anomalies over the latitude bands (a) 60°N-35°N, (b) 20°N-20°S and (c) 35°S-60°S.

 - Figure 8: Again, it would be more helpful to see a comparison with the observations here.

Reply: As the temperature comparison between observations and reananlysis data has been made in previous studies, we cited appropriate papers and added some discussion here. For more details please see the next comment.

 - Line 354: "Large biases in temperature anomalies between two simulations [...] confirming that some of the inhomogeneities seen in ERA-Interim upper stratospheric temperatures [...] have been corrected in ERA5.": How is this statement supported by the figures shown here? The reader can only see that both model runs are different in the stratosphere. Without comparison with the observations, such a statement is not really solid. This problem continues throughout this section.

Reply: We cited recent papers (e.g. Simmons et al., 2020; Marlton et al., 2021) that provide detailed comparison of the stratospheric temperature from ECMWF reanalyses (ERA-Interim and ERA5) with various observational data from radiosonde, satellites and temperature lidars. Simmons et al. (2020) examined the performance of ERA5 using radiosonde and satellite observations. They showed that temperature bias in the upper stratosphere of ERA5 was significantly affected by the addition of the Advanced Microwave Sounding Unit-A (AMSU-A) satellite data between 2000 and 2007 at heights above 15 hPa. Marlton et al. (2021) identified the temperature biases in the upper stratosphere in ECMWF reanalyses (ERA-Interim and ERA5) using a network of temperature lidars. They found a cold bias of −3 ~ −4 K between 10 and 1 hPa in ERA-Interim, while in ERA5 a small bias of magnitude 1 K was found. ERA5 shows a much improved thermal representation of the upper stratosphere up to 1 hPa due to the inclusion of more measurement systems, improved bias correction techniques and model physics, CMIP5 radiative forcings, and a 4Dvar data assimilation system (Hersbach et al., 2020).

Modified texts in the revised manuscript (Lines 412-417) are pasted below: "Large biases in temperature anomalies between two simulations (B_ERA5-A_ERAI) appear in the upper stratosphere for all latitude regions until around 1998, confirming some of the inhomogeneities seen in ERA-Interim upper stratospheric temperatures (Dhomse et al., 2011; McLandress et al., 2014). Some of the recent studies argue that there has been significant improvements in ERA5 temperatures as it

includes more measurements, and uses updated bias correction techniques, model physics and CMIP5 radiative forcings in a 4Dvar data assimilation system (Hersbach et al., 2020; Simmons et al., 2020; Marlton et al., 2021)."

*Related references:*

*Dhomse, S. S., Chipperfield, M. P., Feng, W., and Haigh, J. D.: Solar response in tropical stratospheric ozone: a 3-D chemical transport model study using ERA reanalyses, Atmos. Chem. Phys., 11, 12773–12786, doi:10.5194/acp-11-12773-2011, 2011.*

*Hersbach, H., Bell, B., Berrisford, P., Hirahara, S., Horányi, A., Muñoz-Sabater, J., Nicolas, J., Peubey, C., Radu, R., Schepers, D., Simmons, A., Soci, C., Abdalla, S., Abellan, X., Balsamo, G., Bechtold, P., Biavati, G., Bidlot, J., Bonavita, M., De Chiara, G., Dahlgren, P., Dee, D., Diamantakis, M., Dragani, R., Flemming, J., Forbes, R., Fuentes, M., Geer, A., Haimberger, L., Healy, S., Hogan, R. J., Hólm, E., Janisková, M., Keeley, S., Laloyaux, P., Lopez, P., Lupu, C., Radnoti, G., de Rosnay, P., Rozum, I., Vamborg, F., Villaume, S., and Thépaut, J.-N.: The ERA5 global reanalysis, Q. J. Roy. Meteorol. Soc., 146, 1999-2049, doi:10.1002/qj.3803, 2020.*

*Marlton, G., Charlton-Perez, A., Harrison, G., Polichtchouk, I., Hauchecorne, A., Keckhut, P., Wing, R., Leblanc, T., and Steinbrecht, W.: Using a network of temperature lidars to identify temperature biases in the upper stratosphere in ECMWF reanalyses, Atmos. Chem. Phys., 21, 6079–6092, doi: 10.5194/acp-21-6079-2021, 2021.*

*McLandress, C., Plummer, D. A., and Shepherd, T. G.: Technical Note: A simple procedure for removing temporal discontinuities in ERA-Interim upper stratospheric temperatures for use in nudged chemistry-climate model simulations, Atmos. Chem. Phys., 14, 1547–1555, doi:10.5194/acp-14-1547-2014, 2014.*

*Simmons, A, Soci, C, Nicolas, J, Bell, B, Berrisford, P, Dragani, R, Flemming, J, Haimberger, L, Healy, S, Hersbach, H, Horányi, A, Inness, A, Munoz-Sabater, J, Radu, R, Schepers, D.: Global stratospheric temperature bias and other stratospheric aspects of ERA5 and ERA5.1, ECMWF Technical Memoranda, 859, doi: 10.21957/rcxqfmg0, 2020.*

- Line 370: "Figure 9 shows the PWLT trends for the zonal mean ozone anomalies over the periods 1984-1997 and 1998-2018 obtained from SWOOSH, A_ERAI and B_ERA5 simulations.": I think for these trends the variables $C\_1$ and $C\_2$ have been introduced in equation (1). Please use these variable names here, too.

Reply: As mentioned above, we have used ILT to replace PWLT. The trends for the periods 1984-1997 and 1998-2018 are marked as "Trend1" and "Trend2" in the revised Figure 9, which are consistent with variable names in Section 2.3.

- Line 375: "It is also important to note that much smaller ozone concentrations in this region means larger retrieval errors for satellite measurements that are used in SWOOSH data set.": What are the errors in this region for the SWOOSH data set? How do these errors translate into these trends? Are these errors able to explain the differences to the model? Please provide some numbers from an error analysis here!

Reply: We added some information in the revised manuscript (Lines 433-436): "It is also important to note that much smaller ozone concentrations in this region means larger retrieval errors for satellite measurements that are used in SWOOSH data set. For example, Davis et al. (2016) found that below

100 hPa, HALOE and SAGE III data sets show up to -60% and +20% biases w.r.t. collocated ozonesonde measurements in the tropics."

*Related references:*
*Davis, S. M., Rosenlof, K. H., Hassler, B., Hurst, D. F., Read, W. G., Vömel, H., Selkirk, H., Fujiwara, M., and Damadeo, R.: The Stratospheric Water and Ozone Satellite Homogenized (SWOOSH) database: a long-term database for climate studies, Earth Syst. Sci. Data, 8, 461-490, doi:10.5194/essd-8-461-2016, 2016.*

- Please give the reader some help to read the figures and tell what is meant with "upper stratosphere" or "lower stratosphere". Some pressure ranges would be very helpful here.

Reply: We have added the information about the upper stratosphere "(10-1 hPa)" in the text (Line 345) and lower stratosphere "(147-32 hPa for the mid-latitudes and from 100-32 hPa for the tropics)" (Line 346).

- Line 380: "In the lower stratosphere, both SWOOSH and A_ERAI show negative trends in the tropical and NH extratropical regions, while B_ERA5 shows increasing trends throughout almost the whole extratropical region.": For me, ERAI and ERA5 look very similar in the lower stratosphere, with ERAI having more pronounced features (e.g. 20°N, 40 hPa) compared to ERA5. SWOOSH trends instead look very differently and do not have such kind of a feature at all, which means that ERA5 performs better in this region. However, this is barely discussed here. Further, the very strong trend seen by ERAI in the Antarctic (which does not agree with SWOOSH) is not mentioned here.

Reply: We have updated the ozone profile trends using ILT, and the results are generally consistent with those using PWLT. We also modified the sentences to improve the clarity (Lines 441-447): "In the lowermost stratosphere near 100 hPa, both SWOOSH and A_ERAI show negative trends (SWOOSH being more negative) in the tropical and NH extratropical regions, while B_ERA5 shows weak recovery or negligible trends. Large disagreement between SWOOSH and A_ERAI appears near 40 hPa where A_ERAI shows significant recovery opposite to the decrease in SWOOSH. However, in the SH mid-latitude lower stratosphere, both simulations show opposite recovery trends to the decrease in SWOOSH. Furthermore, the much stronger recovery signal in the Antarctic lower stratosphere in A_ERAI suggests that the agreement of simulation A_ERAI with SWOOSH is hemispherically asymmetric."

- Line 395: "The trends derived using simple Ordinary Least Square (OLS) method are generally in good agreement with those derived from MLR": I am confused here: Are the trends in Figure S3 computed in a different way? If so, this method should be introduced in the methods section, and it should be mentioned just when introducing the figure in the text. At this point, this sentence is out of place.

Reply: In the revised version, we used the same ILT method to plot trend profiles over the different latitude regions, so we deleted this sentence.

- Line 397: "Hence, these results show that ozone trends from B_ERA5 should be considered with care." I do not agree with this statement. I rather would say that both of the simulation results should be considered with care, depending on which region you look at. For the zonally averaged profiles,

ERAI may outperform ERA5, but after seeing Figure 9, I would be very careful with such general statements.

Reply:  Yes. We agree and sorry for the confusion. It was intended to suggest that latest reanalysis data from ECMWF (ERA5) is better than ERA-Interim in some respects but is not free from inhomogeneities, hence ERA5 based trend estimates should be treated carefully. More detailed modifications are marked in the revised manuscript.

Section 3.3

 - As mentioned in the general part, I would suggest to remove this section and prepare it for a different publication. In that case, the observational data could be introduced properly (which is missing now completely), and additional, such as satellite data sets of AoA could be considered in a more detailed comparison. This would also help to reduce the number of figures in this paper: 12 figures are really a lot and I fear that the reader will lose its focus at this point.

- If the authors decide to keep this section, they should carefully restructure the whole paper, in order to connect the ozone trends with the AoA trends. This should also be reflected by the title, the introduction and the conclusions. Right now, these parts are rather separate.

Reply: Thank you. We decide to keep this AoA discussion because AoA is a main diagnostic that helps to understand the differences in ozone. We tried to improve flow and clarity of this part in the manuscript and at the same time make sure the paper as concise as possible. Some information related to AoA has been added in the title, Abstract, Introduction, Section 2.2 and Section 4. Details are marked in the revised manuscript.

Section 4:

 - Line 504: "The PWLT-based regression model shows that both SWOOSH and A_ERAI show negative trends since 1998 in the NH extratropical lower stratosphere where, in contrast, B_ERA5 shows increasing trends.": As mentioned above, I do not see the picture that clear as the authors do. From the figures shown in this manuscript, I would be more careful with such statements.

Reply: We modified this sentence in Lines 584-588: "The ILT-based regression model shows that although simulation A_ERAI is consistent with SWOOSH with negative trends in the lowermost stratosphere in the tropical and NH mid-latitude regions since 1998, there still exist large differences with more significant positive trends seen in A_ERAI at ~40 hPa in the NH extra-tropics and in the Antarctic, while B_ERA5 shows inconsistent increasing trends in both NH and SH mid-latitude regions. Hence, trends derived using either simulation should be considered with care."

 - Line 516: "Our results show that although B_ERA5 shows better agreement with observed TCO than A_ERAI, they do not confirm that B_ERA5, based on the newer reanalyses, performs better in simulating stratospheric ozone overall.": Again, I think this broad statement is not backed by the findings of this paper.

Reply: We modified this sentence (Lines 598-603): "Our results show that although simulation B_ERA5 shows better agreement with observed TCO anomalies compared to A_ERAI, there still exist larger biases over certain regions (e.g. NH in winter). Similarly, although simulation A_ERAI is

more consistent with SWOOSH with negative trends in the lowermost stratosphere in the tropical and NH mid-latitude regions post-1998, there also exist larger biases over certain regions (e.g. ~40 hPa in the NH extra-tropics). With the newer reanalysis, B_ERA5 does not perform better in simulating stratospheric ozone overall, and both simulations should be treated carefully for trend estimates."

PS (other modifications):
We use SWOOSH (v2.7) to replace the original v2.6. The differences are mainly in the lower stratosphere where the new v2.7data shows more negative trends in both periods 1984-1997 and 1998-2020.
Some related references are cited and updated in the revised manuscript.

---

## Author Comment (AC2)

**Reply to Reviewer #2:**

We thank the reviewer for his/her comments and suggestions. These comments are given below in black text, followed by our responses in blue text.

This paper reports on ozone results from TOMCAT chemistry-transport model (CTM) runs using two different meteorological ECMWF reanalyses as input, ERA5 and its predecessor ERAI. While agreement of model runs driven by ERA5 appear to agree better with total ozone observations, biases with respect to observations are larger in ozone profile data for ERA5 driven TOMCAT than for ERAI driven. In addition recent ozone trends in total columns and stratospheric ozone differ depending on the reanalysis data used. Their main conclusion is that the current ERA5 reanalysis is not able to reproduce in CTMs the observed ozone changes, in particular, in the lower stratosphere.

The topic of this paper is very relevant and is within the scope of ACP. Recent stratospheric ozone changes are governed by changes in ozone-depleting substances (chemical contribution) and circulation/transport (dynamical contribution), the latter strongly influenced by changes in greenhouse gases. In particular, the differences in the reanalyses mainly affect the circulation pattern in the model and ozone transport. Both CTMs and trend regression models applied to observations rely on input from meteorological analyses (forcing and proxies) in order to separate the dynamical and chemical part of the overall ozone trends. Uncertainties in the reanalyses therefore can affect trend estimates derived from CTMs and, possibly, observations.

I recommend publication in ACP after addressing the following points.

l.138: The sentence "Sofieva et al. and Steinbrecht et al. ..." does not belong here as they report on profile trends not total column trends.

Reply: Sorry for this error. We have removed the sentence and provided a web link for the validation document (https://datastore.copernicus-climate.eu/documents/satellite-ozone/C3S2_312a_Lot2_PQAR_O3_latest.pdf).

l.281: The table caption is confusing (what is the meaning of "… based on ..."). What is shown here are the correlations (r2) between the regression model and the data timeseries.

Reply: Sorry for this confusion. We have removed the table and added the information of correlations (r2) to Figure 4 in the revised manuscript as suggested by Reviewer #1.

Eq. 1 (and Fig. 5): I would suggest to use ILT (independent linear trends) rather than PWLT. PWLT are quite sensitive to the turning point (time of maximum of stratospheric halogen content) being quite close to the Pinatubo period. It could be that differences between the models and observations could become more pronounced with the ILT approach.

Reply: We thank the reviewer for this insightful comment and suggestion. In the revised manuscript, we have used independent linear trends (ILT) to replace PWLT. We have rewritten the regression methods in Section 2.3. The updated results for TCO trends and ozone profile trends for the models and observations are compared and shown in Figure 4, Figure 9 and Figure S4. We find the results are generally consistent with those using PWLT. Some detailed modifications are shown in red text in the revised manuscript.

[Figure]

Figure 4: Peak contributions (in %) from piecewise linear trend and explanatory variable terms (see equation (1)) to the total ozone column variability during DJF and JJA for (a, b) 60°N-35°N, (c, d) 20°N-20°S and (e, f) 35°S-60°S for C3S, A_ERAI and B_ERA5 during 1979-2018. Error bars indicate the confidence bounds at the 95% statistical significance level quantified by ± 2 standard deviations (σ). The determination coefficients (R-squared) of the regression model for DJF and JJA mean TCO time series from C3S, A_ERAI and B_ERA5 over the 60°N-35°N, 20°N-20°S and 35°S-60°S regions are presented in each plot.

[Figure]

Figure 9: Latitude-pressure cross sections of the piecewise linear trends of ozone anomalies (%/decade) over the periods 1984-1997 and 1998-2018 for (a, b) SWOOSH, (c, d) A_ERAI and (e, f) B_ERA5, respectively. Stippled regions indicate where the trends are statistically significant at 95% level of confidence.

[Figure]

Figure S4: Vertical profile of linear trends in ozone (%/decade) from SWOOSH (black solid line), A_ERAI (blue dashed line) and B_ERA5 (red dash-dot line) over the periods (a-c) 1984-1997 and (d-f) 1998-2018. Results are for 60°N-35°N, 20°N-20°S and 35°S-60°S zonal regions. Error bars show standard deviations at 2σ.

l. 362: here the large differences in temperatures between A_ERAI and B_ERA5 before 1998 is seen as a proof that ERA5 upper stratospheric temperatures are improved (due to some corrections in the assimilation of MLS). Has this been validated? A comparison of ERAI/ERA5 with MLS or other temperatures could show this.

Reply: The upper stratospheric temperatures are improved by ERA5, this has been validated in previous studies: Simmons et al. (2020) examined the performance of ERA5 using radiosonde and satellite observations. They showed that temperature bias in the upper stratosphere of ERA5 was significantly affected by the addition of the Advanced Microwave Sounding Unit-A (AMSU-A) satellite data between 2000 and 2007 at heights above 15 hPa. Marlton et al. (2021) identified the temperature biases in the upper stratosphere in ECMWF reanalyses (ERA-Interim and ERA5) using a network of temperature lidars. They found a cold bias of −3 ~ −4 K between 10 and 1 hPa in ERA-Interim, while in ERA5 a small bias of magnitude 1 K was found. ERA5 shows a much improved thermal representation of the upper stratosphere up to 1 hPa due to the inclusion of more measurement systems (e.g. the AMSU-A and the Constellation Observing System for Meteorology, Ionosphere, and Climate (COSMIC)), improved bias correction techniques and model physics, CMIP5 radiative forcings, and a 4Dvar data assimilation system (Hersbach et al., 2020).

We have modified this sentence and also cited some references (e.g. Simmons et al., 2020; Marlton et al., 2021) in the revised manuscript (Lines 412-417): "Large biases in temperature anomalies between two simulations (B_ERA5-A_ERAI) appear in the upper stratosphere for all latitude regions until around 1998, confirming that some of the inhomogeneities seen in ERA-Interim upper stratospheric temperatures (Dhomse et al., 2011; McLandress et al., 2014). Some recent studies argue that there has been significant improvements in ERA5 temperatures as it includes more measurements, and uses updated bias correction techniques and model physics, CMIP5 radiative forcings in a 4Dvar data assimilation system (Hersbach et al., 2020; Simmons et al., 2020; Marlton et al., 2021)."

*Related references:*

*Hersbach, H., Bell, B., Berrisford, P., Hirahara, S., Horányi, A., Muñoz-Sabater, J., Nicolas, J., Peubey, C., Radu, R., Schepers, D., Simmons, A., Soci, C., Abdalla, S., Abellan, X., Balsamo, G., Bechtold, P., Biavati, G., Bidlot, J., Bonavita, M., De Chiara, G., Dahlgren, P., Dee, D., Diamantakis, M., Dragani, R., Flemming, J., Forbes, R., Fuentes, M., Geer, A., Haimberger, L., Healy, S., Hogan, R. J., Hólm, E., Janisková, M., Keeley, S., Laloyaux, P., Lopez, P., Lupu, C., Radnoti, G., de Rosnay, P., Rozum, I., Vamborg, F., Villaume, S., and Thépaut, J.-N.: The ERA5 global reanalysis, Q. J. Roy. Meteorol. Soc., 146, 1999-2049, doi:10.1002/qj.3803, 2020.*

*Marlton, G., Charlton-Perez, A., Harrison, G., Polichtchouk, I., Hauchecorne, A., Keckhut, P., Wing, R., Leblanc, T., and Steinbrecht, W.: Using a network of temperature lidars to identify temperature biases in the upper stratosphere in ECMWF reanalyses, Atmos. Chem. Phys., 21, 6079–6092, doi:10.5194/acp-21-6079-2021, 2021.*

*Simmons, A, Soci, C, Nicolas, J, Bell, B, Berrisford, P, Dragani, R, Flemming, J, Haimberger, L, Healy, S, Hersbach, H, Horányi, A, Inness, A, Munoz-Sabater, J, Radu, R, Schepers, D.: Global stratospheric temperature bias and other stratospheric aspects of ERA5 and ERA5.1, ECMWF Technical Memoranda, 859, doi: 10.21957/rcxqfmg0, 2020.*

l.368: "Thus, the differences in the upper stratospheric temperatures from the reanalysis data sets drive the differences in ozone anomalies in this region." Please explain why this is the case, e.g. cooler (warmer) temperatures produce more (less) ozone.

Reply: Thank you. We have added the explanation in the manuscript (Lines 420-422): "the differences in the upper stratospheric temperatures from the reanalysis data sets drive the differences

in ozone anomalies in this region, as cooler (warmer) temperatures causes more (less) ozone when photochemical processes dominate (e.g. Stolarski et al., 2012)."

*Related references:*

*Stolarski, R. S., Douglass, A. R., Remsberg, E. E., Livesey, N. J., Gille, J. C.: Ozone temperature correlations in the upper stratosphere as a measure of chlorine content, J. Geophys. Res., 117, D10305, doi:10.1029/2012JD017456., 2012.*

l.482: "The increasing AoA in B_ERA5 after 1998 as well as the older age in the NH lower stratosphere, suggest that other transport pathways (such as downward transport/reduced transport in the troposphere) might have been responsible for the increasing ozone in the NH extratropical lower stratosphere in B_ERA5". It should be mentioned that the aging of AoA in the NH appears, however, consistent with SWOOSH (observational) trends and the notion of reduced downward transport. Could this mean that there is some model issue here (uncertainties in transport patterns in the model). Please discuss.

Reply: Yes. To make the discussion more complete, we have added some information to the revised manuscript (Lines 546-554): "A possible explanation might be changes in the vertical resolution (and changes in number and type of observations) assimilated in ERA5. For example, in NH mid-high latitude, B_ERA5 shows somewhat older AoA in the lowermost stratosphere (near 100 hPa), but between 70 to 10 hPa, B_ERA5 shows slightly younger AoA compared to A_ERAI. In contrast, B_ERA5 minus A_ERAI ozone differences seen in Figure 7 (a, b) remain positive throughout the stratosphere with an exception of slightly negative values near 10 hPa. This clearly shows that changes in vertical transport can lead to larger changes in lower stratospheric ozone as ozone lifetime increases exponentially from a few days near middle stratosphere to a few years in the lower stratosphere. A similar feature is observed in the SH mid-latitudes. However, at SH high-latitudes, B_ERA5 shows somewhat positive AoA compared to A_ERAI but simulated ozone differences are negative in the lower-middle stratosphere."

---

## Author Response (AR2)

Response to minor corrections:

Line/figure numbers refer to the track changes document:

Line 181: Still a typo in "Steinbrecht"
Line 540: "CLaMs" -> "CLaMS"

Reply: corrected.

Line/figure numbers refer to the supplement:

Figure S1: In the caption, TCO are introduced, but it should be SCO (as mentioned in the title and in the author's response).

Reply: corrected.